# Regorafenib inhibits EphA2 phosphorylation and leads to liver damage via the ERK/MDM2/p53 axis

Hao Yan [1], Wentong Wu[1], Yuhuai Hu[2,3], Jinjin Li[1], Jiangxin Xu[1], Xueqin Chen[4,5], Zhifei Xu [1], Xiaochun Yang[1], Bo Yang[6], Qiaojun He[1,2] & Peihua Luo [1,7,8] ✉

The hepatotoxicity of regorafenib is one of the most noteworthy concerns for patients, however the mechanism is poorly understood. Hence, there is a lack of effective intervention strategies. Here, by comparing the target with sorafenib, we show that regorafenib-induced liver injury is mainly due to its nontherapeutic target Eph receptor A2 (EphA2). EphA2 deficiency attenuated liver damage and cell apoptosis under regorafenib treatment in male mice. Mechanistically, regorafenib inhibits EphA2 Ser897 phosphorylation and reduces ubiquitination of p53 by altering the intracellular localization of mouse double minute 2 (MDM2) by affecting the extracellular signal-regulated kinase (ERK)/MDM2 axis. Meanwhile, we found that schisandrin C, which can upregulate the phosphorylation of EphA2 at Ser897 also has protective effect against the toxicity in vivo. Collectively, our findings identify the inhibition of EphA2 Ser897 phosphorylation as a key cause of regorafenib-induced hepatotoxicity, and chemical activation of EphA2 Ser897 represents a potential therapeutic strategy to prevent regorafenib-induced hepatotoxicity.

Regorafenib, an oral small molecule kinase inhibitor (SMKI), possesses multikinase repression regulation related to tumor proliferation, angiogenesis, and the tumor microenvironment[1]. In the first two years, regorafenib was approved by the Food and Drug Administration (FDA) as the first line for the treatment for refractory metastatic colorectal cancer patients and advanced gastrointestinal stromal tumor patients[2]. The RESORCE study showed that regorafenib provides a significant and clinically meaningful improvement in overall survival in patients with hepatocellular carcinoma progressing during sorafenib treatment. This finding was associated with an increase in median

survival from 7.8 months to 10.6 months[3–5]. Thus, regorafenib was approved by the FDA as a second-line treatment for hepatocellular carcinoma patients[6]. However, severe and fatal hepatotoxicity was labeled as a black box warning by the FDA as soon as it was put on the market. Randomized placebo-controlled trials (CORRECT, GRID, RESORCE and CONCUR) showed that 39–70% of patients suffered elevated levels of alanine transaminase (ALT), and 58–93% of patients suffered elevated levels of aspartate transaminase (AST)[7,8]. Meanwhile, CORRECT trials indicated that 3.2% (3/93) of patients presented severe toxic hepatitis, and even 1.1% (1/93) had fatal hepatotoxicity after

[1]Center for Drug Safety Evaluation and Research of Zhejiang University, College of Pharmaceutical Sciences, Zhejiang University, Hangzhou 310058, China. [2]Innovation Institute for Artificial Intelligence in Medicine of Zhejiang University, Hangzhou 310018, China. [3]Laboratory of Fruit Quality Biology/Zhejiang Provincial Key Laboratory of Horticultural Plant Integrative Biology/The State Agriculture Ministry Laboratory of Horticultural Plant Growth, Development and Quality Improvement, Zhejiang University, Hangzhou 310058, China. [4]Department of Oncology, Affiliated Hangzhou Cancer Hospital, Zhejiang University School of Medicine, Key Laboratory of Clinical Cancer Pharmacology and Toxicology Research of Zhejiang Province, Hangzhou 310002, China. [5]Cancer Center, Zhejiang University, Hangzhou 310058, China. [6]Institute of Pharmacology & Toxicology, College of Pharmaceutical Sciences, Zhejiang University, Hangzhou 310058, China. [7]Department of Pharmacology and Toxicology, Hangzhou Institute of Innovative Medicine, College of Pharmaceutical Sciences, Zhejiang University, Hangzhou 310018, China. [8]Key Laboratory of Clinical Cancer Pharmacology and Toxicology Research of Zhejiang Province, Affiliated Hangzhou Cancer Hospital, Zhejiang University School of Medicine, Hangzhou 310002, China. ✉e-mail: peihualuo@zju.edu.cn

regorafenib administration[9]. The mechanism of hepatotoxicity caused by regorafenib is still obscure with a lack of effective intervention strategies. It is a very clear unmet clinical need to be solved.

With the deepening of the understanding of p53, it has been found to be involved in the regulation of various life activities[10]. There is accumulating evidence that p53 plays an important regulatory role in liver pathogenesis[11–14]. For example, chronic ethanol feeding in rats has been shown to increase the hepatic mRNA abundance, acetylation, and transcriptional activity of p53, and further enhance apoptosis[15]. Additionally, hepatocyte apoptosis was linked to p53 activation in mice fed a high fat diet and the p53 inhibitor pifithrin-α p-nitro diminished hepatic triglyceride accumulation and lipotoxicity[16]. In recent studies, p53 was also confirmed to participate in drug-induced toxicity, but the role of p53 was still unknown. In acetaminophen-induced liver injury, p53 plays a protective role by regulating the drug-metabolizing enzymes and transporters cytochrome P450s (CYPs), sulfo-transferases (SULTs) and mitochondrial ribosomal proteins (MRPs)[17], while in cisplatin-induced nephrotoxicity, p53 may mediate tubular cell apoptosis through its regulation of gene transcription[18]. Based on the abovementioned study, p53 represents a strong and controversial regulator of biological activities. Studying the role of p53 in managing drug-induced toxicity is still a fascinating topic.

Recent investigations have shown that EphA2 has many important and diverse biological functions[19,20]. EphA2 was found to be expressed in the lens fiber cell layer, and both EphA2 mutations in humans and EphA2 inactivation in mice result in cataracts[21,22]. EphA2 is also expressed at high levels in the developing kidney. Miao et al. reported that EphA2 negatively regulated hepatocyte growth factor-induced branch morphogenesis of Madin-Darby canine kidney cells[23,24]. Although the liver highly expresses EphA2, the role of EphA2 in maintaining hepatic function is still obscure. Recent studies have also reported that EphA2 may behave as an injury-and stress-responsive regulator. Phosphorylation at Ser897 and Tyr588 plays an important role in the migration and repair of endothelial cells during endothelial injury. In the damaged region, the decrease in the phosphorylation at Tyr588 and increase in phosphorylation at Ser897 promote cell motility and proliferation[24]. In our study, we found that the phosphorylation of EphA2 at Ser897 plays an important role in liver homeostasis and further study was performed based on the phosphorylation of EphA2.

In this study, we revealed a mechanism for regorafenib-induced hepatotoxicity, wherein regorafenib upregulates p53 levels by removing the ubiquitin and then causes mitochondria-dependent apoptosis in hepatocytes. In addition, we confirmed that the phosphorylation of EphA2 at Ser897 plays an important role in the stability of p53 by regulating the extracellular signal-regulated kinase (ERK)/MDM2 axis and we found that EphA2 inhibitors, which can inhibit Ser897 phosphorylation have the same regulatory effect on p53 as regorafenib. Finally, we demonstrate that recovering EphA2 Ser897 phosphorylation by schisandrin C is a potential therapeutic strategy for regorafenib-induced hepatotoxicity.

## Results

### Regorafenib causes apoptosis and mitochondrial dysfunction in hepatocytes

First, we established an animal model to simulate the hepatotoxicity caused by regorafenib in clinical use. Male C57BL/6J mice were treated with 200 mg/kg/day or 400 mg/kg/day regorafenib for 3 weeks by gavage. The ALT and AST levels showed no significant change after treatment with 200 mg/kg/day regorafenib for 3 weeks but slightly increased with 400 mg/kg/day (Supplementary Fig. 1a). Then, we used 400 mg/kg/day regorafenib for the rest of the animal experiments. We prolonged the period of treatment to 6 weeks and found that mice had yellowish livers compared to the control. H&E staining showed nuclear shrinkage (Fig. 1a), which is consistent with the results of liver biopsy of regorafenib-induced hepatotoxicity in patients[25]. The ALT and AST

levels, indicators of liver injury, also increased after regorafenib treatment (Fig. 1b). Considering the potential difference between sexes in toxicological assessment and the relevance to the clinical situation, we then evaluated the effect of regorafenib on male and female mice for 6 weeks. The livers of both male and female mice showed similar changes in appearance after regorafenib treatment (Supplementary Fig. 1b). Structural disorder and hepatocyte damage were observed in both sexes by H&E staining (Supplementary Fig. 1c). The levels of ALT and AST slightly increased at the end of the third week and considerably changed during the sixth week. However, there was no significant difference between male mice and female mice (Supplementary Fig. 1d). The abovementioned data show that gender may not be a factor for the sensitivity of regorafenib-induced hepatotoxicity in mice.

Furthermore, we validated the toxic effect of regorafenib in human hepatocytes. The survival rates of human primary hepatocytes were significantly reduced upon regorafenib treatment (Fig. 1c), which was consistent in the human hepatocyte cell line HL-7702 (Supplementary Fig. 2a). Apoptosis is one of the pathogenic characteristics of hepatotoxicity and it has been reported that regorafenib can induce apoptosis in several cancer cells and human hepatic cells[26–31]. Thus, Annexin V-PI staining followed by flow cytometry was performed on regorafenib-treated cells to examine hepatocyte apoptosis. As a result, regorafenib overtly elicited an apoptotic response in hepatocytes (Fig. 1d). Meanwhile, the western blot analysis of the expression level of cleaved poly (ADP-ribose) polymerase (c-PARP), an endogenous marker of apoptosis, corroborated our findings (Supplementary Fig. 2b). Z-VAD-FMK, a pan-caspase inhibitor, effectively protected hepatocytes against regorafenib-induced survival rate reduction (Supplementary Fig. 2c) and apoptosis (Supplementary Fig. 2d, e). The change in mitochondrial membrane potential (MMP) is an early event of apoptosis[32] and we then detected the MMP by JC-1 staining followed by flow cytometry. The obtained results showed that regorafenib decreased the MMP in a dose- and time-dependent manner (Fig. 1e). The loss of MMP is related to mitochondrial dysfunction, and as previously reported, regorafenib can inhibit the respiratory chain and increase the mitochondrial production of superoxide which would further influence the MMP[30,31]. Thus, we also used MitoTracker and ATP measurements to detect the effect of regorafenib on mitochondria. Regorafenib decreased mitochondrial mass (Fig. 1f) and ATP production in a dose-dependent manner (Fig. 1g) which further confirmed that regorafenib-induced mitochondrial dysfunction. Moreover, TdT-mediated dUTP nick-end labeling (TUNEL) staining (Fig. 1h) and western blot analysis showed that regorafenib-induced liver apoptosis in vivo (Fig. 1i). Taken together, our findings showed that regorafenib caused hepatotoxicity by inducing hepatocyte mitochondrial dysfunction and apoptosis.

### The expression of p53 is upregulated with regorafenib-induced hepatotoxicity in vivo and in vitro

Upregulation of p53 levels is one of the main factors of apoptosis caused by mitochondrial damage[33,34]. We first examined the expression of p53 in liver tissues. The western blot results showed that the protein level of p53 was much higher in regorafenib-treated mice than in normal controls (Fig. 2a), and the upregulated expression level of p53 was further confirmed by immunohistochemistry (IHC) analysis (Supplementary Fig. 3a). We next evaluated the protein level of p53 in vitro. Notably, the expression of p53 was significantly increased in both primary human hepatocytes and HL-7702 cells after regorafenib treatment (Fig. 2b and Supplementary Fig. 3b). Immunofluorescence analysis and western blot analysis showed that regorafenib increased p53 levels in both the cytoplasm and nucleus (Supplementary Fig. 3c, d). Considering the role of p53 in regulating apoptosis and cell fate[35,36], we constructed p53-deficient hepatocytes using two siRNAs targeting p53 to assess whether regorafenib-induced p53 upregulation was essential to

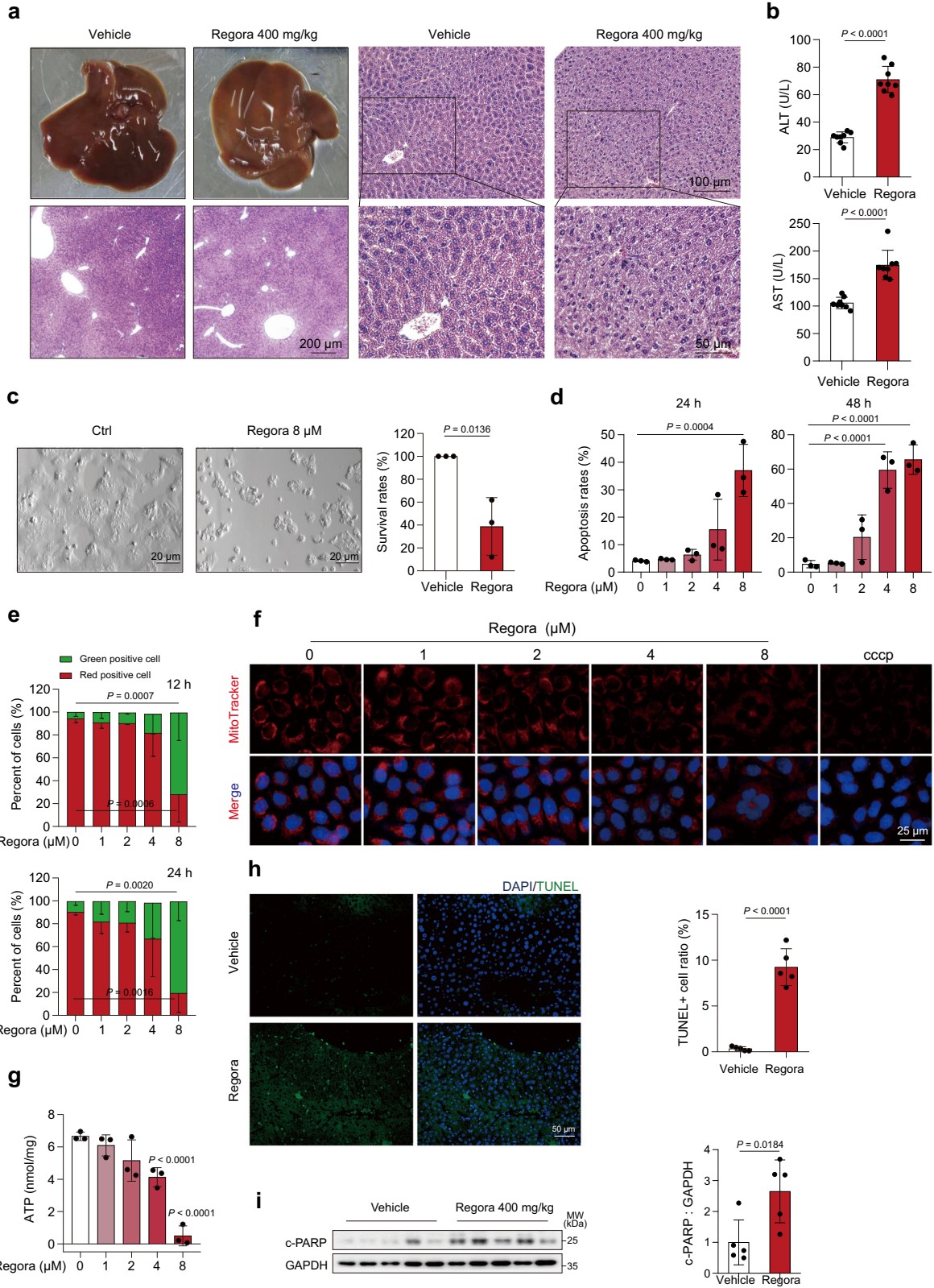

apoptosis and mitochondrial dysfunction in hepatocytes. p53-deficient hepatocytes showed higher survival rates (Fig. 2c), less cleavage of PARP (Fig. 2d) and lower apoptosis rates (Fig. 2e). Meanwhile, JC-1 staining followed by flow cytometry showed that MMP greatly recovered in p53-deficient hepatocytes compared with negative controls after regorafenib treatment (Fig. 2f). Silencing of p53 could also alleviate regorafenib-induced loss of mitochondrial mass (Fig. 2g) and ATP

production inhibition (Fig. 2h). Collectively, these results suggested that the increased expression of p53 was involved in regorafenib-induced hepatocyte apoptosis and mitochondrial dysfunction.

## Regorafenib stabilized p53 by inhibiting its ubiquitination

Next, we sought to clarify the mechanism for the p53 increase. First, we investigated whether regorafenib increased the transcriptional

**Fig. 1 | Regorafenib causes hepatocyte apoptosis and mitochondrial dysfunction. a, b** C57BL/6J male mice were treated with 0.5% CMC-Na or 400 mg/kg/day regorafenib for 6 weeks (n = 8 per group). **a** Representative image of liver and representative images of H&E staining. For 40× magnification, scale bar: 200 μm; 100× magnification, scale bar: 100 μm; for 200× magnification, scale bar: 50 μm. **b** The levels of serum ALT and AST were analyzed (n = 8 per group). **c** HPH from 3 donors were treated with DMSO or 8 μM regorafenib for 48 h. Morphologic changes of HPH were observed by optical light microscope. Scale bar: 20 μm. **d** The apoptosis rates of HL-7702 cells treated with regorafenib for 24 h or 48 h were measured by flow cytometry analysis with Annexin V-PI staining. n = 3 independent experiments. e HL-7702 cells were treated with for 12 h or 24 h, and MMP was

detected by flow cytometry with JC-1 staining. n = 3 independent experiments. **f, g** HL-7702 cells were treated with 0, 1, 2, 4 or 8 μM regorafenib for 24 h. n = 3 independent experiments. Mitochondrial mass was detected by MitoTracker staining (red). Representative images are shown. **g** The concentration of ATP were measured. **h** Representative images of TUNEL staining (green) in liver tissues. Scale bar: 50 μm. **i** The expression level of c-PARP in liver tissues was analyzed by western blot (n = 5 biologically independent samples per group). Data were expressed as mean ± SD. Unpaired two-sided Student's *t* test for **b**, **c**, **h** and **i** or one way ANOVA followed by Tukey post hoc test for **d**, **e** and **g**. Source data are provided as a Source Data file. Regora regorafenib, HPH human primary hepatocytes, MMP mitochondrial membrane potential, MW molecular weight.

level of the *TP53* gene. Quantitative real-time polymerase chain reaction (qRT–PCR) analysis showed no significant change in the mRNA level of p53 in regorafenib-treated HL-7702 cells and mouse liver (Fig. 2i). We then applied cycloheximide (CHX), an inhibitor of protein synthesis, to examine the effect of regorafenib on the turnover of p53 protein. Western blot analysis showed that regorafenib prolonged the half-life of p53, suggesting that increased p53 protein levels are caused by the inhibition of its degradation (Fig. 2j). It has been reported that p53 can be degraded by both the ubiquitin-proteasome pathway and the autophagy–lysosome pathway and the ubiquitin–proteasome system is the major regulated pathway of p53 degradation[37,38]. MG132, a proteasome inhibitor, did not further increase the expression of p53 after regorafenib treatment, suggesting that regorafenib-induced p53 accumulation was associated with ubiquitin–proteasome pathway inhibition (Fig. 2k). Moreover, immunoprecipitation assay revealed that the ubiquitination of p53 was greatly inhibited by regorafenib (Fig. 2l). As previously reported, regorafenib could also induce lethal autophagy arrest by stabilizing specific proteins in glioblastoma. In this study, we found that LC3-I/II levels decreased after treatment with 8 μM regorafenib in hepatocytes which suggested that regorafenib inhibited the initiation of autophagy (Supplementary Fig. 4a). However, chloroquine (CQ), a late-stage autophagy inhibitor, did not further increase p53 expression both in CQ treatment and combination treatment (Supplementary Fig. 4b). The obtained results suggested that autophagy was not involved in regulating p53 stability under normal circumstances and regorafenib treatment. In summary, these data confirm our hypothesis that regorafenib-induced liver injury is caused by inhibiting the ubiquitination of p53.

## EphA2 is the key target of regorafenib-induced liver injury

The target of the drug is an important factor in the development of toxic effects; therefore, we next focused on the target of regorafenib. Regorafenib is a derivative of sorafenib and the F atom on phenoxy is the only difference between them (Fig. 3a) while the degree of hepatotoxicity is different during clinical use. We confirmed that regorafenib had stronger inhibitory effects on survival rates in HL-7702 cells at the same multiples of Cmax (Supplementary Fig. 5a), which was consistent with the clinical observations. Meanwhile, sorafenib failed to increase the expression level of p53 in a dose and time-dependent manner (Supplementary Fig. 5b). Therefore, we reasoned that the mechanism of regorafenib-induced hepatotoxicity may be unmasked by comparing it with sorafenib. According to the reference[39] and the information in DrugBank, we found that, in addition to known therapeutic targets such as vascular endothelial growth factor receptor (VEGFR), platelet derived growth factor receptor (PDGFR), Raf-1 proto-oncogene, serine/threonine kinase (RAF-1) and B-Raf proto-oncogene, serine/threonine kinase (BRAF), regorafenib also inhibits discoidin domain receptor tyrosine kinase 2 (DDR2), neurotrophic receptor tyrosine kinase 1 (TrkA), EphA2, mitogen-activated protein kinase 11 (SAPK2), and PTK5 protein tyrosine kinase 5 (PTK5) (Fig. 3a). To gain insight into the role of these differential targets in regorafenib-induced hepatotoxicity, we treated HL-7702 cells with scramble siRNA or

specific siRNAs, and then assessed survival rates for each treatment. Intriguingly, only the siRNA targeting EphA2 could greatly rescue regorafenib-induced hepatocyte death (Supplementary Fig. 5c and Fig. 3b). Western blot analysis showed that knocking down EphA2 could not only attenuated c-PARP elevation but also recovered the p53 level to normal (Fig. 3c). Next, we detected the effect of EphA2 on regorafenib-induced hepatocyte apoptosis and mitochondrial dysfunction. Annexin V-PI staining followed by flow cytometry showed that knockdown of EphA2 alleviated HL-7702 cell apoptosis caused by regorafenib treatment (Fig. 3d). The MMP was also improved in the EphA2 knockdown hepatocytes by JC-1 staining analysis (Fig. 3e), as well as the loss of mitochondrial mass (Fig. 3f) and ATP production dysfunction (Fig. 3g). Furthermore, based on these findings, we indicated that EphA2 may be the key target in regorafenib-induced hepatotoxicity, which mediates hepatocyte apoptosis by regulating p53 levels.

To investigate the effect of EphA2 on regorafenib-induced liver injury in vivo, mice were injected with AAV8-TBG-control (wild-type) or AAV8-TBG-sh*EphA2* (hepatocyte-specific EphA2 knockdown). The knockdown efficiency EphA2 was detected by immunofluorescence analysis (Supplementary Fig. 6). *EphA2* knockdown reversed elevated levels of both ALT and AST induced by regorafenib treatment (Fig. 3h), and liver pathology injury (Fig. 3i, upper panel). These results suggested that EphA2 plays an important role in regorafenib-induced hepatotoxicity. Moreover, the level of p53 was detected by IHC staining and western blot analysis (Fig. 3i, middle panel and j). These data suggested that knocking down EphA2 could effectively reduce regorafenib-induced p53 accumulation. In addition, the apoptosis level of the liver caused by regorafenib was also alleviated in *EphA2* knockdown mice (Fig. 3i, lower panel). Taken together, these data demonstrated the role of EphA2, the nontherapeutic target of regorafenib, in the upregulation of p53 and in regorafenib-induced hepatocyte apoptosis.

## The Inhibition of EphA2^Ser897 phosphorylation is related to p53 upregulation

As a tyrosine kinase, whether the phosphorylation level of EphA2 is involved in the regulation of p53 homeostasis needs to be considered. According to the dependence on ligand binding and tyrosine kinase activity, the EphA2 signaling can be defined as canonical or noncanonical pathway. The EphA2 canonical signaling pathway depends on the binding of EFNA1 (also known as ephrin A1), the most preferred ligand of EphA2. Autophosphorylation at Tyr588 (pY-EphA2) occurs in response to EFNA1 binding, and promotes the tyrosine kinase activity and activation of downstream signaling[24]. Meanwhile, the EphA2 noncanonical pathway is mainly regulated via phosphorylation at Ser897 (pS-EphA2) in the linker region, which is independent of ligand binding.

First, we detected whether the canonical pathway and pY-EphA2 play a role in regorafenib-induced hepatotoxicity. The obtained results showed that pY-EphA2 was decreased under regorafenib treatment (Supplementary Fig. 7a); however, mimicking EphA2^Tyr588 dephosphorylation by mutating EphA2 tyrosine 588 to alanine (EphA2-

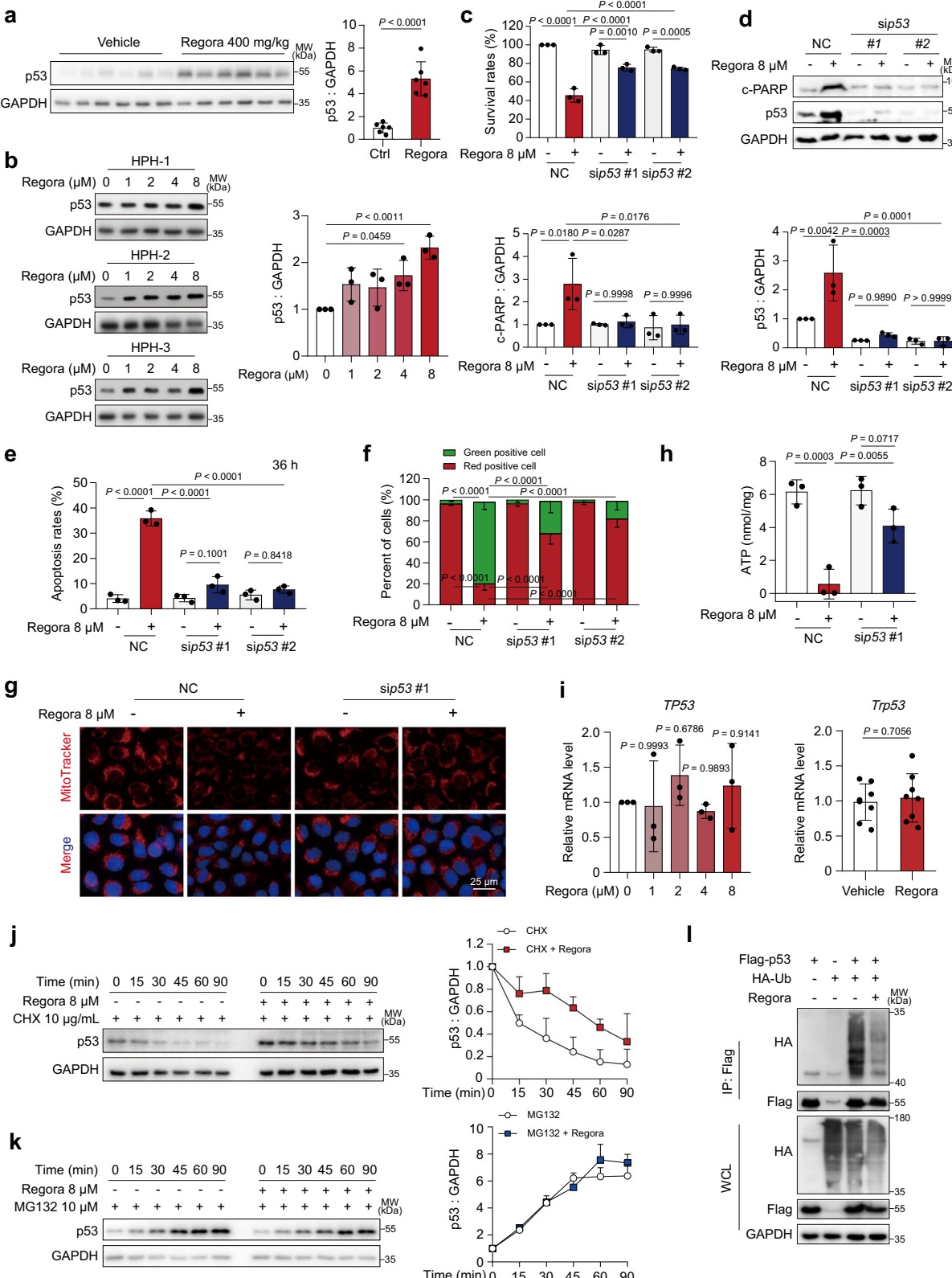

Y588A) could not induce HL-7702 cell apoptosis (Supplementary Fig. 7b) or mitochondrial dysfunction (Supplementary Fig. 7c). These data suggested that pY-EphA2 may not be involved in regorafenib-regulated hepatocyte homeostasis disturbance. Furthermore, we also detected the impact of the classical ligand EFNA1 on the proposed pathway. We noticed that with regorafenib treatment alone, the expression of EFNA1 was greatly reduced, which may explain the effect of regorafenib on the inhibition of Tyr588 phosphorylation (Supplementary Fig. 8a). Silencing or overexpression of EFNA1 could not reverse regorafenib's effect on the level of p53 (Supplementary Fig. 8b, c). After treatment with EFNA1 recombinant protein, Tyr588 was phosphorylated, which was consistent with the reported reference[40,41]. However, there was little influence on the survival rate (Supplementary Fig. 8d) or the expression of p53 (Supplementary Fig. 8e). Because

**Fig. 2 | Regorafenib upregulates the expression of p53 both in vivo and in vitro.**
**a** The expression level of p53 in liver tissues from Fig. 1a was analyzed by western blot ($n = 6$ per group). **b** HPH from 3 donors were treated with regorafenib as indicated for 24 h. The expression level of p53 was detect by western blot. NC or p53 siRNAs transfected HL-7702 cells were treated with or without 8 μM regorafenib for 36 h (for survival rate detection (**c**) and apoptosis rate measurement (**e**)) or 24 h (for protein expression level (**d**) and MMP measurement (**f**)). $n = 3$ independent experiments. **g, h** NC transfected or p53-knockdown HL-7702 cells were treated as indicated for 24 h. $n = 3$ independent experiments. **g** The mitochondrial mass was detected by MitoTracker staining (red). Representative images are shown. **h** The concentration of ATP were measured. **i** The mRNA expression levels of *TPS3* in HL-7702 cells treated with regorafenib for 24 h ($n = 3$ per group) and *Trp53* in liver tissues from mice ($n = 8$ per group) were detected. **j** HL-7702 cells were treated with regorafenib for 6 h and CHX were addded before harvest for the indicated time period. The expression level of p53 was detected by western blot. $n = 3$ independent experiments. **k** HL-7702 cells were treated with MG132 or/and regorafenib as indicated. The expression level of p53 was detected by western blot. $n = 2$ independent experiments. **l** The cell lysates of HEK293T with different treatment were subjected to immunoprecipitation by using Flag-beads, followed by western blot to detect HA-tag or Flag-tag expression. Blots are representative of three independent experiments. Data were expressed as mean ± SD. Unpaired two-sided Student's $t$ test for **a** and **i** right or one way ANOVA followed by Tukey post hoc test (**b, c, e–h** and **i** left). Source data are provided as a Source Data file. Regora regorafenib, HPH human primary hepatocytes, NC negative control, MMP mitochondrial membrane potential, MW molecular weight.

tyrosine kinase activity is crucial for the EphA2 canonical signaling pathway, we also applied the EphA2 loss-of-function mutation K646M[42] to test whether EphA2 tyrosine kinase activity was required for stabilizing p53. When we overexpressed EphA2-K646M, regorafenib could still increase the level of p53 (Supplementary Fig. 9). Thus, the abovementioned data suggested that regorafenib-induced hepatotoxicity was highly unlikely to occur by targeting the canonical signaling pathway.

Next, we studied the role of pS-EphA2 in regorafenib-induced hepatotoxicity. We analyzed the sequence among different species and found that Ser897/898 was highly conserved (Fig. 4a). After regorafenib treatment, the phosphorylation of EphA2 at Ser898 in mouse livers was downregulated, preliminarily suggesting a relationship between pS-EphA2 inhibition and hepatotoxicity (Fig. 4b). We then sought to determine the effect of EphA2[Ser897] phosphorylation on regorafenib-induced liver injury. Immunofluorescence analysis showed that regorafenib reduced pS-EphA2 on the membrane with little influence on the EphA2 level (Supplementary Fig. 10a, b), which was confirmed by western blotting (Fig. 4c). As the dose and treatment period increased, the levels of pS-EphA2 and p53 presented the opposite tendency. To further investigate the role of pS-EphA2 in regorafenib-induced hepatotoxicity, we mutated EphA2 serine 897 to alanine or aspartic acid, which could mimic phosphorylation inactivation (EphA2-S897A) or continuous phosphorylation activation (EphA2-S897D). We noted that a robust and high expression level of p53 was observed after transfecting the EphA2-S897A plasmid compared with the vector. Reciprocally, EphA2-S897D significantly reduced the level of p53 (Fig. 4d). Similar results were obtained by immunofluorescence analysis (Fig. 4e). Furthermore, compared with the control group, mimicking EphA2 inactivation significantly prolonged the half-life of p53, while continuous phosphorylation activation accelerated the degradation of p53 even without regorafenib treatment (Fig. 4f). To this extent, we found that pS-EphA2 is involved in the regulation of p53 stability. Because of the direct changes in the expression of p53, we also detected the effect of pS-EphA2 activation or inactivation on cell survival. SRB analysis showed that overexpression of EphA2-S897A significantly promoted cell death alone (Supplementary Fig. 11a). Moreover, EphA2-S897A could also induce hepatocyte apoptosis and mitochondrial dysfunction, while EphA2-S897D had almost no effect compared with the vector (Supplementary Fig. 11b, c). Moreover, under regorafenib treatment, EphA2-S897A transfection further increased regorafenib-induced p53 upregulation (Fig. 4g), while EphA2-S897D attenuated p53 expression by the opposite effect (Fig. 4h). These results suggested that the phosphorylation of EphA2 at Ser897 is vital to sustain the physiological regulation of hepatocytes.

In conclusion, these results demonstrated that the phosphorylation inhibition of EphA2 at Ser897 rather than Tyr588 was the main cause of regorafenib-induced liver injury by stabilizing p53, and EphA2-S897A overexpression could increase the level of p53 directly in vitro.

## EphA2[Ser897] phosphorylation regulates p53 by the ERK/MDM2 axis

We then investigated how pS-EphA2 regulates the proteasomal pathway degradation of p53. MDM2 is a p53-specific E3 ubiquitin ligase, and its activated form can transport p53 into the cytoplasm and degrade p53 through the proteasome pathway, which plays an important role in the regulation of p53 stability[43]. Ser166 is close to the nuclear localization sequence region of MDM2 and is reported to be essential for transporting p53 out of the nucleus for degradation[44]. Considering that phosphorylation regulation is quite close to the activation of EphA2, we first examined the phosphorylation level of MDM2[Ser166] (pS-MDM2) after regorafenib treatment. The obtained results showed that regorafenib could reduce pS-MDM2 in a time- and dose-dependent manner with little effect on the prototype (Fig. 5a). Similar results were obtained in liver tissues by western blot analysis (Fig. 5b). To validate the effect of Ser166 phosphorylation on the localization of MDM2, we separated the cytoplasm and nucleus and detected the distribution of MDM2. The obtained results showed that the nuclear localization of MDM2 was significantly reduced after regorafenib treatment in both hepatocytes and liver tissues (Fig. 5c). We then examined whether stimulating EphA2[Ser897] phosphorylation activation/inactivation could affect the phosphorylation of MDM2 and the intracellular distribution of MDM2 in the same way as regorafenib treatment. Western blot analysis showed that overexpression of EphA2-S897A could reduce pS-MDM2, while EphA2-S897D increased pS-MDM2 levels (Fig. 5d). As a result, the distribution of MDM2 was considered to be changed due to the regulation of MDM2 phosphorylation. Immunofluorescence analysis showed that overexpression of EphA2-S897A decreased the nuclear localization of MDM2, whereas EphA2-S897D accumulated MDM2 in nuclei (Fig. 5e). Moreover, overexpression of EphA2-S897D improved the nuclear distribution of MDM2, which was inhibited by regorafenib (Supplementary Fig. 12). Based on these findings, we speculated that the drug that could inhibit pS-EphA2 may cause liver injury by upregulating the p53 level. To validate these hypotheses, we examined the effect of an EphA2 inhibitor ALW-II-41-27 and a potential EphA2 inhibitor among FDA-approved drugs on p53 expression levels. Surprisingly, those drugs that failed to inhibit pS-EphA2 had no regulatory effect on pS-MDM2 and p53 (Supplementary Fig. 13). Only ALW-II-41-27 and bosutinib, which could induce an increase in p53, accompanied the decreased phosphorylation of EphA2[Ser897] and MDM2[Ser166] (Fig. 5f). From the abovementioned data, we could construct a correlation between pS-EphA2 and pS-MDM2; however, the regulatory mechanism of EphA2 and MDM2 is still elusive.

As previously reported, the AKT-mTOR and ERK pathways could be activated through ligand-independent activation of EphA2 (phosphorylation of Ser897)[45]. It has been reported that the phosphorylation of AKT serine/threonine kinase (AKT) at Ser473 (pS-AKT) and ERK at Tyr204 (pY-ERK) can catalyse the phosphorylation of MDM2 at Ser166, and the influence of pY-ERK overwhelms that of pS-AKT in liver tissue[46]. Considering this, we detected the phosphorylation of AKT and ERK and found that pY-ERK was reduced following

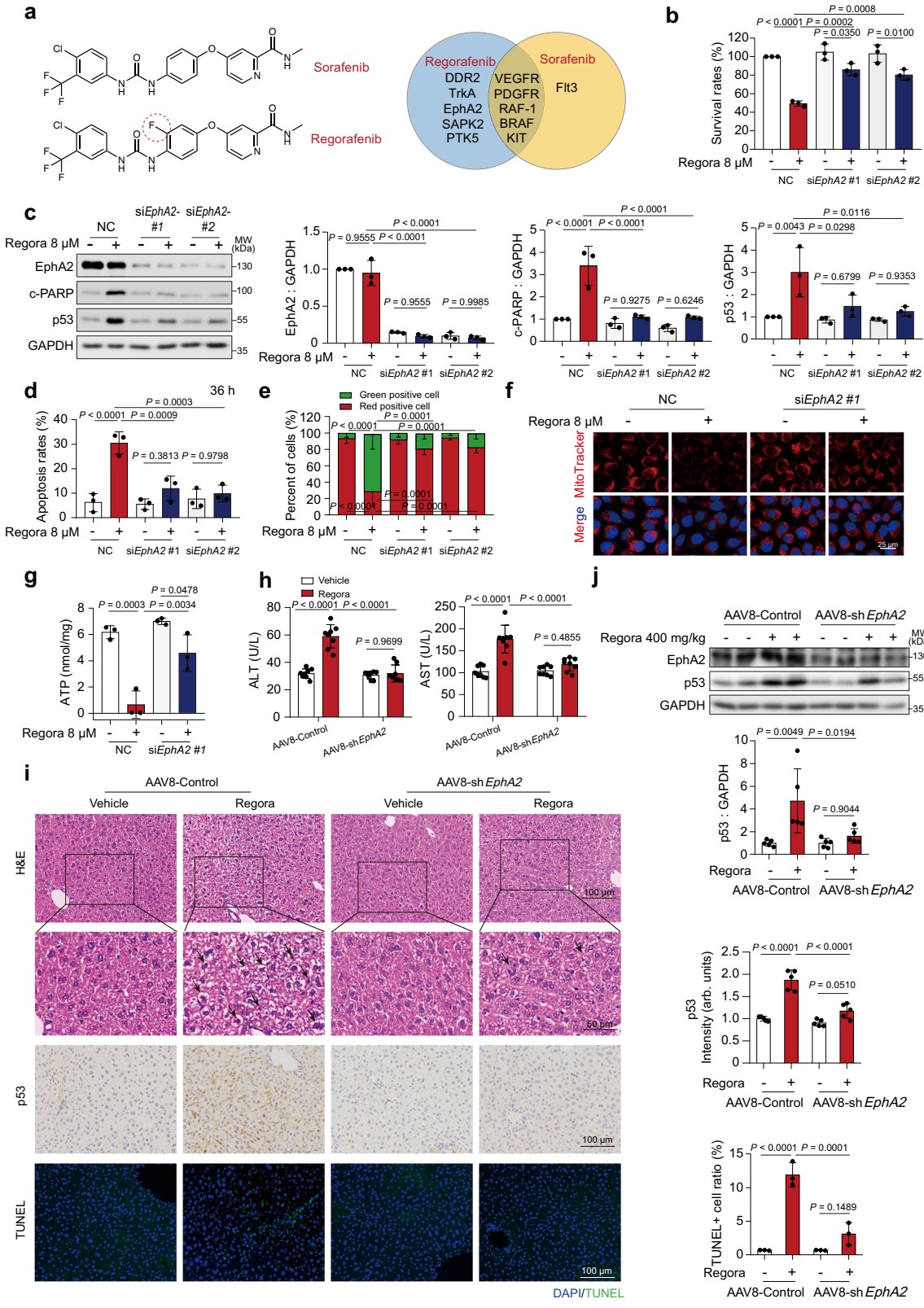

regorafenib treatment accompanied by the inhibition of pS-EphA2 and pS-MDM2 (Fig. 5g and Supplementary Fig. 14a), while regorafenib failed to influence pS-AKT (Supplementary Fig. 14b). Furthermore, immunofluorescence analysis also showed that the increase in MDM2 cytoplasmic localization was accompanied by the inhibition of pS-EphA2 and pY-ERK (Supplementary Fig. 15a). Then, we treated HL-7702 cells with EphA2-S897A or EphA2-S897D

plasmids and assessed the role of pS-EphA2 in regulating ERK and MDM2. Immunofluorescence analysis showed that the overexpression of EphA2-S897A decreased pY-ERK and promoted the localization of MDM2 in the cytoplasm, whereas EphA2-S897D had the exact opposite result (Supplementary Fig. 15b). Collectively, these data confirm our hypothesis that the inhibition of pS-EphA2 may increase p53 levels through the ERK/MDM2 axis.

**Fig. 3 | EphA2 is the key target of regorafenib-induced liver injury. a** Structural and target comparison of regorafenib and sorafenib. NC or EphA2 siRNAs transfected HL-7702 cells were treated with or without 8 μM regorafenib for 36 h (for survival rate (**b**) and apoptosis rate measurement (**d**)) or 24 h (for expression level (**c**) and MMP measurement (**e**)). $n = 3$ independent experiments. **f, g** NC transfected or EphA2-knockdown HL-7702 cells were treated with or without 8 μM regorafenib for 24 h. $n = 3$ independent experiments. **f** The mitochondrial mass in HL-7702 cells was stained by MitoTracker (red). Representative images are shown. **g** The concentration of ATP in HL-7702 cells were measured by ATP detection kit. **h–j** AAV8-TBG-Control or AAV8-TBG-sh*EphA2* were injected into C57BL/6J male mice through tail vein. After three weeks, C57BL/6J mice were fed with 0.5% CMC-Na or 400 mg/kg/day regorafenib for another 6 weeks ($n = 8$ per group). **h** The levels of serum ALT

and AST were analyzed ($n = 8$ per group). **i** Representative images of H&E staining, immunohistochemistry for p53 staining and TUNEL staining in liver tissues from C57BL/6J mice. Black arrowheads indicated nuclear shrinkage or structure disorder with vacuolization in specific region. For 100× magnification, scale bar: 100 μm; for 200× magnification, scale bar: 50 μm. The immunohistochemistry for p53 staining was quantified by densitometric analysis. One area from five mice per group. **j** The expression levels of EphA2 and p53 in liver tissues were analyzed by western blot. $n = 3$ independent experiments. Data were expressed as mean ± SD. One way ANOVA followed by Tukey post hoc test for **b–e**, **g–j**. Source data are provided as a Source Data file. Regora regorafenib, NC negative control, MMP mitochondrial membrane potential, MW molecular weight.

The noncanonical activation of EphA2 is independent of ligand, and RSK, AKT, PKA, ERK, etc., are reported to regulate the phosphorylation of EphA2 S897 depending on different stimuli[40]. To determine the relationship between EphA2 and ERK under regorafenib treatment, we first treated hepatocytes with regorafenib for a period of time. Due to obvious inhibition of phosphorylation of EphA2 at S897 in a short period of time, we shortened the actuation duration of regorafenib and tried to distinguish the changes. From the obtained data, we determined that p-EphA2 inhibition was much more sensitive to regorafenib stimuli than p-ERK, both in response time and degree of inhibition (Fig. 5h). Furthermore, when EphA2 was knocked down, the effect of p-ERK inhibition could be greatly alleviated, which suggested that EphA2 was upstream of ERK under regorafenib treatment (Supplementary Fig. 16a). We also applied the ERK activator TBHQ to the treatment with regorafenib[47]. As shown in Supplementary Fig. 16b, TBHQ could activate ERK and slightly increase the level of p-EphA2 (Ser897), while TBHQ could not recover the inhibition of p-EphA2 when combined with regorafenib. Regorafenib could still inhibit the phosphorylation of EphA2 and reduce the level of p-ERK with TBHQ treatment. Thus, we confirmed that EphA2 is upstream of ERK, at least in regorafenib-treated hepatocytes.

### Hepatocyte-specific overexpression of EphA2-S898A induced hepatocyte apoptosis by increasing p53 levels in vivo

Next, we constructed a liver-specific EphA2-S898A overexpression mouse model (EphA2 Ser897 was identified as Ser898 in mice) by AAV8 virus injection to further confirm the effect of EphA2 in vivo. As expected, EphA2$^{Ser898}$ dephosphorylation mimicked the upregulation of ALT and AST in mice (Fig. 6a). At the same time, EphA2-S898A-overexpressing mouse livers had severe nuclear shrinkage and immune cell infiltration in H&E staining (Fig. 6b) and increased hepatocyte apoptosis (Fig. 6c). Furthermore, we investigated the levels of p53 and MDM2 phosphorylation. From the results of IHC analysis, we observed that the mice injected with the AAV8-TBG-EphA2-S898A virus had an increased level of p53 accompanied by attenuated MDM2 phosphorylation (Fig. 6d). Similarly, the upregulation of p53 and c-PARP was negatively associated with the level of p-MDM2, which was detected by western blot analysis (Fig. 6e). Furthermore, the downregulated pS-MDM2 resulted in MDM2 retention in the cytoplasm of the liver of AAV8-TBG-EphA2-S898A virus-treated mice (Fig. 6f).

The abovementioned data demonstrated that the phosphorylation level of EphA2 at Ser898 is critical for liver tissue homeostasis, and the low phosphorylation of Ser898 directly induces p53-dependent apoptosis in liver tissue.

### Recovering EphA2$^{Ser897}$ phosphorylation by genetic or pharmacological approaches rescued regorafenib-induced hepatotoxicity

Based on our findings above, we speculated that inhibition of EphA2$^{Ser897}$ phosphorylation was the initial cause for regorafenib-induced hepatotoxicity and that recovering the phosphorylation of

EphA2$^{Ser897}$ could rescue the hepatotoxicity of regorafenib. Therefore, we specifically overexpressed EphA2 or EphA2-S898D in mice by injecting mice with AAV8 carrying the TBG promoter with EphA2 or EphA2-S898D. Three weeks later, the mice were treated with 0.5% CMC-Na or 400 mg/kg/day regorafenib (Fig. 7a). Interestingly, under regorafenib treatment, overexpression of EphA2 increased the levels of ALT and AST, while EphA2-S898D partially recovered the increased levels of ALT and AST compared with the vehicle group after regorafenib treatment (Fig. 7b). These data suggested that the overexpression of EphA2 was still affected by regorafenib and amplified the inhibitory effect of regorafenib. Histopathology of liver sections showed that EphA2-S898D-overexpressing mice had considerably improved liver histology with markedly reduced nuclear shrinkage caused by regorafenib treatment (Fig. 7c). TUNEL staining also revealed that hepatocyte apoptosis was alleviated in regorafenib-treated mice by overexpressing EphA2-S898D (Fig. 7d). Furthermore, we detected the levels of p53 and pS-MDM2 in those mouse models. IHC analysis showed that EphA2-overexpressing mice further attenuated pS-MDM2 and increased the level of p53 compared with vehicle mice after regorafenib treatment, while the expression of EphA2-S898D partially recovered the level of pS-MDM2 and p53 (Fig. 7e, f). Taken together, these results demonstrated that regorafenib caused hepatocyte apoptosis by inhibiting EphA2 phosphorylation at Ser897, which could be a potential therapeutic or preventive target for regorafenib-induced hepatotoxicity.

A total of 41 approved drugs and 45 natural products were screened (Supplementary Fig. 17a, b), and we noted that aripiprazole, schisandrin C and dioscin could significantly induce pS-EphA2 (Supplementary Fig. 17c). Furthermore, the survival rate was detected when combined with regorafenib (Supplementary Fig. 17d). The most effective compound, schisandrin C, was identified. Meanwhile, it could also decrease p53 and upregulate pY-ERK and pS-MDM2 (Fig. 8a). Schisandrin C is a dibenzocyclooctadiene lignan isolated from the fruits of Schisandra chinensis (Turcz.) Baill and is reported to have a good protective effect on drug-induced liver injury[48]. Here, we speculated that combining schisandrin C could alleviate regorafenib-induced hepatotoxicity by stimulating pS-EphA2. As shown in Supplementary Fig. 18a, the addition of schisandrin C alleviated hepatocyte apoptosis caused by regorafenib. JC-1 staining showed that schisandrin C attenuated the decreased MMP (Supplementary Fig. 18b), loss of mitochondrial mass (Supplementary Fig. 18c) and ATP production dysfunction (Supplementary Fig. 18d), suggesting that schisandrin C could greatly ameliorate regorafenib-induced mitochondrial dysfunction in hepatocytes. Moreover, the inhibitory effect of regorafenib on the nuclear localization of MDM2 was partially improved by schisandrin C (Supplementary Fig. 19a), and western blot analysis showed that schisandrin C could alleviate regorafenib-induced pS-EphA2 inhibition, pY-ERK inhibition and p53 increase (Supplementary Fig. 19b). We also detected whether the effect of schisandrin C on p53 was dependent on EphA2 kinase activity. When EphA2 and EphA2-K646M were overexpressed, schisandrin C could still downregulate the level of p53, suggesting that schisandrin C regulated p53

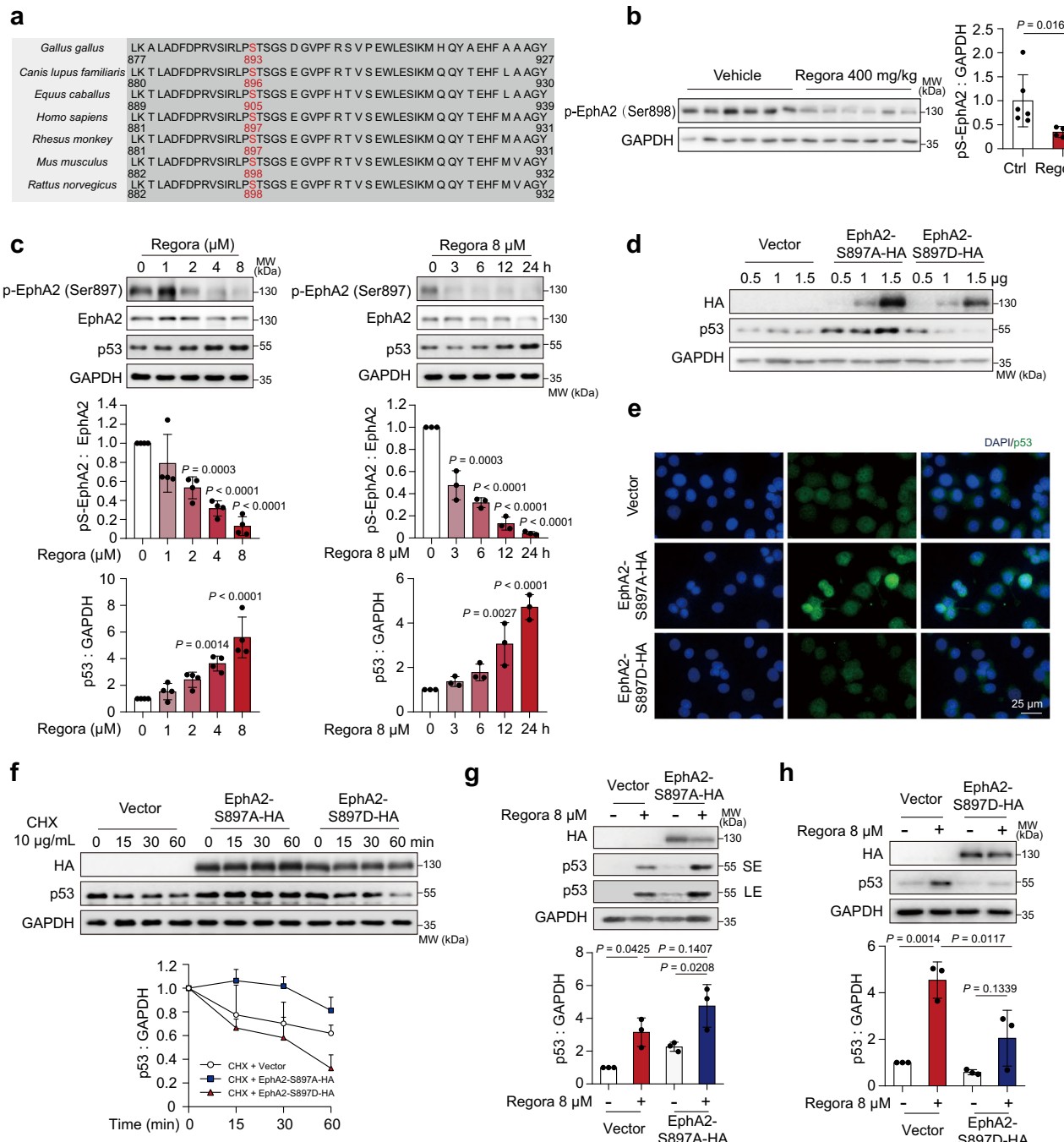

**Fig. 4 | Phosphorylation of EphA2^Ser897 inhibition is associated with p53 upregulation. a** The partial EphA2 sequence among different species. **b** C57BL/6J mice were treated with 0.5% CMC-Na or 400 mg/kg/day regorafenib for 6 weeks. The expression level of p-EphA2 (Ser898) in liver tissues was analyzed by western blot (n = 6 per group). **c** HL-7702 cells were treated with 0, 1, 2, 4 or 8 μM regorafenib for 24 h (left, n = 4 independent experiments) or treated with 8 μM regorafenib for 0, 3, 6, 12 or 24 h (right, n = 3 independent experiments). The expression levels of p-EphA2 (Ser897), EphA2 and p53 were detected by western blot. **d, e** HL-7702 cells were transfected with 0.5, 1.0 or 1.5 μg vector, EphA2 S897A plasmid or EphA2 S897D plasmid for 24 h. **d** The expression levels of HA and p53 were analyzed by western blot. Blots are representative of three independent experiments. **e** The distribution and expression of p53 in HL-7702 cells were observed by immunofluorescence. Representative images are shown from three independent experiments. Scale bar: 25 μm. **f** HL-7702 cells transfected with 1.0 μg EphA2 S897A plasmid or EphA2 S897D plasmid were treated with 10 μg/mL CHX for 0, 15, 30 or 60 min. The expression levels of HA and p53 were analyzed by western blot. n = 3 independent experiments. **g, h** HL-7702 cells were transfected with 1.0 μg EphA2 S897A plasmid or EphA2 S897D plasmid for 24 h before treatment with 8 μM regorafenib for 24 h. Western blot was used to detect the expression level of HA and p53. n = 3 independent experiments. Data were expressed as mean ± SD. Unpaired two-sided Student's t test for **c** or one way ANOVA followed by Tukey post hoc test for **g** and **h**. Source data are provided as a Source Data file. Regora regorafenib, MW molecular weight, SE short exposure, LE long exposure.

stability through the noncanonical EphA2 signaling pathway (Supplementary Fig. 19c). All above mentioned results confirmed the protective role of schisandrin C in regorafenib-induced hepatotoxicity by regulating the level of pS-EphA2 in vitro. Furthermore, we sought to determine whether schisandrin C has therapeutic potential in vivo by treating mice with regorafenib and/or schisandrin C. As shown in Fig. 8b, the levels of serum ALT and AST were greatly decreased after combining schisandrin C with regorafenib compared to regorafenib

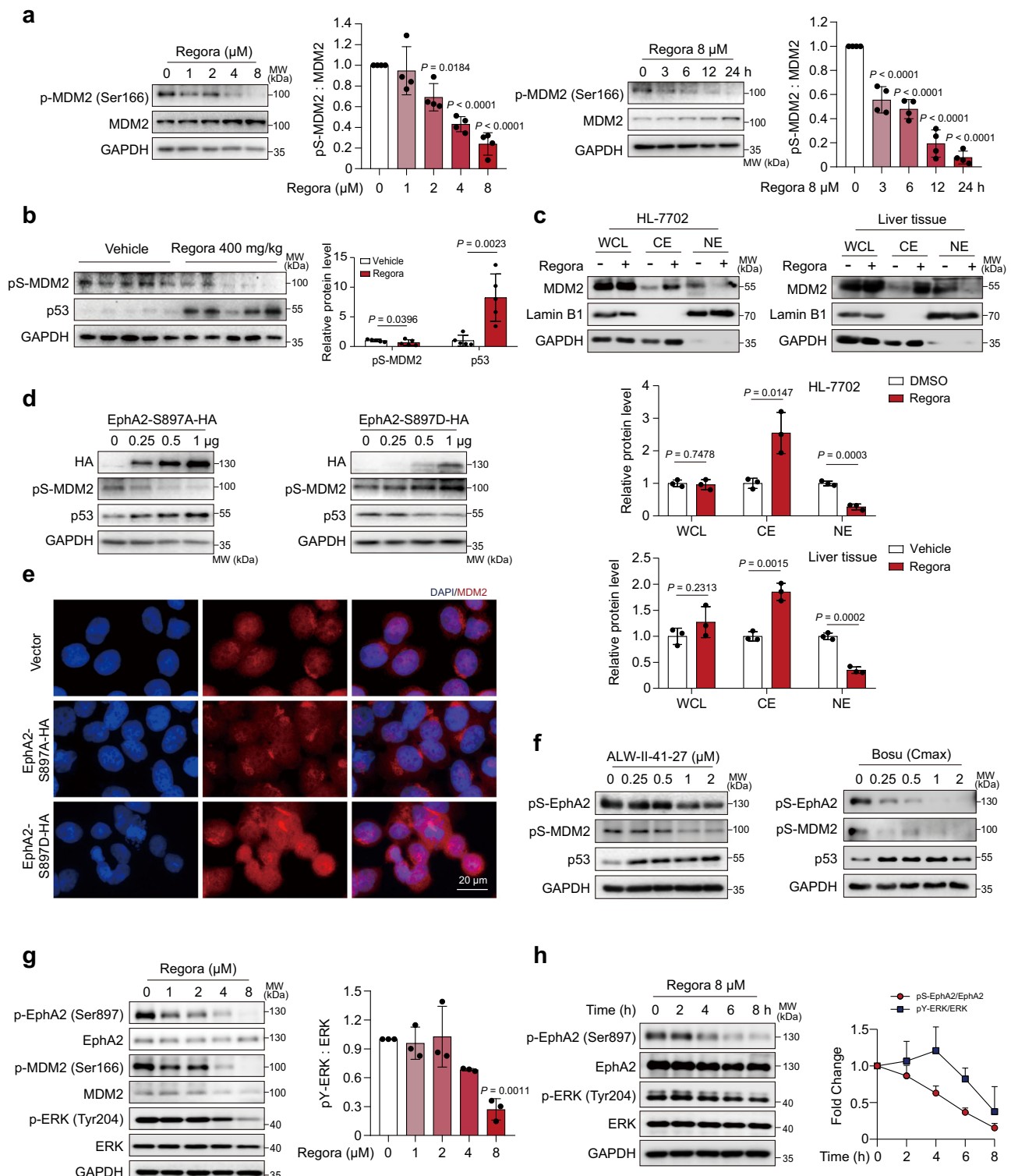

treatment alone, indicating the protective effect of schisandrin C. The increased LW/BW ratio (ratio of liver weight to body weight) was also abrogated by schisandrin C (Supplementary Fig. 20). Accordingly, schisandrin C improved liver histology with less nuclear shrinkage (Fig. 8c). Meanwhile, TUNEL staining revealed that schisandrin C alleviated hepatocyte apoptosis after regorafenib treatment (Fig. 8d). IHC analysis showed that the abnormal expression of pY-ERK, pS-MDM2 and p53 caused by regorafenib treatment could be improved by schisandrin C (Fig. 8e), suggesting that the protective role of schisandrin C is relative to the ERK/MDM2/p53 axis. The expression of pS-EphA2 and

p53 was further confirmed by western blot analysis (Fig. 8f). To verify the effect of schisandrin C on the anticancer activity of regorafenib, we detected the effect of the combination of schisandrin C and regorafenib in two hepatocellular carcinoma cell lines, Bel-7402 and Hep-G2, and two colon cancer cell lines, HCT-116 and SW-480. The obtained results suggested that schisandrin C failed to affect the anticancer effect of regorafenib (Supplementary Fig. 21). Taken together, our results demonstrated that schisandrin C could alleviate the liver injury caused by regorafenib treatment, suggesting that recovering EphA2[Ser897] phosphorylation by schisandrin C could be a potential

**Fig. 5 | The phosphorylation of EphA2^Ser897 regulates p53 by ERK/MDM2 axis.**
**a** HL-7702 cells were treated with regorafenib as indicated. The expression levels of p-MDM2 (Ser166) and MDM2 were analyzed by western blot. $n = 4$ independent experiments. **b** C57BL/6J mice were treated with or without 400 mg/kg/day regorafenib for 6 weeks. The expression levels of p-MDM2 (Ser166) and p53 in liver tissues were analyzed by western blot ($n = 5$ per group). **c** HL-7702 cells treated with 8 μM regorafenib for 24 h ($n = 3$ independent experiments) or liver tissues ($n = 3$ per group) were used to detect the expression level of Lamin B1 and MDM2 in whole cell, cytoplasm and nuclei by western blot. **d**, **e** HL-7702 cells transfected with vector, EphA2 S897A plasmid or EphA2 S897D plasmid as indicated for 24 h. **d** The expression levels of HA, p-MDM2 (Ser166) and p53 were analyzed by western blot. **e** The localization of MDM2 was observed by immunofluorescence. Representative images are shown from three independent experiments. Scale bar: 20 μm. **f** HL-

7702 cells were treated with ALW-II-41-27 or bosutinib as indicated for 24 h. Related proteins were analyzed by western blot. Blots are representative of three independent experiments. **g** HL-7702 cells were treated with regorafenib as indicated for 24 h. The expression levels of related proteins were analyzed by western blot. The relative level of p-ERK (Tyr204) was analyzed by densitometric analysis. $n = 3$ independent experiments. **h** HL-7702 cells were treated with regorafenib as indicated. The expression levels of related proteins were analyzed by western blot. The fold change of pS-EphA2/EphA2 and pY-ERK/ERK were analyzed by densitometric analysis. $n = 3$ independent experiments. Data were expressed as mean ± SD. Unpaired two-sided Student's $t$ test for **b** and **c** or one way ANOVA followed by Tukey post hoc test for **a** and **g**. Source data are provided as a Source Data file. Regora regorafenib, MW molecular weight, WCL whole cell lysate, CE cytoplasmic extraction, NE nuclear extraction.

preventive and therapeutic approach for hepatotoxicity caused by regorafenib in the clinic.

## Discussion

Drug-induced liver injury (DILI) is the second most common cause of drug attrition during development as well as for postmarketing withdrawal[49–51]. DILI can manifest as various forms of acute and chronic liver diseases and can be fatal to patients without effective intervention. In this study, we demonstrated a previously undefined role for EphA2 (nontherapeutic target of regorafenib) inhibition as the main cause of regorafenib-induced hepatotoxicity and discovered the molecular process in p53 stability with EphA2 inhibition. We identified EphA2 Ser897, identified as Ser898 in mice, as the key phosphorylated residue required for proteasome pathway degradation of p53 by regulating the ERK/MDM2 axis. In addition, we applied several in vitro and in vivo approaches to confirm that recovering EphA2 Ser897 phosphorylation-mediated p53 degradation by schisandrin C is a promising therapeutic strategy for regorafenib-induced hepatotoxicity, which would benefit the safe application of regorafenib in the clinic (Fig. 9).

Currently, research on drug-induced side effects is mainly focused on the negative effects caused by drug therapeutic targets, which may explain the toxicity of a few drugs. EGFR is required for the growth and development of the epidermis, and it is generally believed that the cutaneous toxicity caused by EGFR inhibitors is closely related to the inhibition of EGFR[52], although the exact regulatory mechanism has not been found. VEGFR inhibitor-induced hypertension is mainly caused by the inhibition of VEGFR, which is involved in regulating angiogenesis[53]. Immune disorders caused by immunotherapy are also thought of as direct suppression of immune targets, such as colitis caused by PD-1 inhibitors[54]. In addition, the effect of noncanonical targets on toxicity production has rarely been studied.

Regorafenib is derived from sorafenib and has a small structural modification. The difference between them is only one F-atom on phenoxy; however, there is a significant difference in hepatotoxicity in the clinic[55]. It has been reported that the incidence of all grades of ALT/AST elevation was 21–25% under sorafenib treatment with no fatal cases of hepatic failure, while the incidence was 45–65% for all grades and 6% for grade 3-4 in regorafenib administration and is sufficiently high and severe to acquire a boxed label warning[56]. Consistent with this notion, we found that sorafenib induced lower hepatocyte apoptosis than regorafenib in hepatocytes and failed to affect the expression of p53. We discovered that regorafenib-induced hepatotoxicity originated from the nontherapeutic target EphA2 by comparing the different targets of regorafenib and sorafenib. No histologic toxicities were detected in the liver by delivering EphA2-targeting siRNA, suggesting that the basal level of EphA2 may not be related to the physiological function of the liver[57]. To date, the role of EphA2 in regulating the pathological function of the liver with stimuli challenge is still undefined. In this study, we discovered that inhibiting the phosphorylation of EphA2 at Ser897 is vital for the pathological and

physiological changes in the liver, mainly manifesting in regulating hepatocyte survival. We also found a function of EphA2 in managing the stability of p53, enriching the theory of p53 homeostasis to some extent. In Figs. 2c and 3b, the survival rate under regorafenib treatment was greatly improved with EphA2 knockdown compared to the effect of p53 knockdown, which suggests that EphA2 acts as an upstream factor in regulating cell death. A similar phenomenon was observed by JC-1 staining (Figs. 2f and 3e). EphA2, as a kinase receptor, has been reported to regulate several downstream signaling pathways in response to different stimuli. Apart from p53, there could be other EphA2-regulated proteins determining cell fate that merit further investigation.

Considering the EphA2 inhibition in maintaining liver homeostasis, we also detected other SMKIs that identified EphA2 as a nontherapeutic target and further confirmed the relationship between EphA2 Ser897 phosphorylation and p53 stability (Fig. 5f and Supplementary Fig. 13). Detection of EphA2 phosphorylation at Ser897 would be beneficial for potential hepatotoxicity screening at the preclinical stage. Moreover, in drug design, it should be carefully considered to avoid the inhibitory activity of EphA2 at Ser897.

Under different circumstances, EphA2 plays various or even opposite roles in regulating normal tissue function. EphA2 is closely associated with maintaining the morphology, structure and differentiation of lens epithelial cells and is crucial for lens clarity with age[21]. However, EphA2 contributes to brain blood barrier damage and neuronal death after ischemic stroke. The brain tissue of EphA2^−/− mice had lower levels of proapoptotic proteins and higher levels of antiapoptotic proteins than wild-type mice following stroke[58]. In a study of atherosclerosis, depletion of endothelial EphA2 attenuated progression to advanced atherosclerotic plaques by reducing early monocyte recruitment to the plaque[59]. Similarly, genetic ablation of EphA2 protects against bleomycin-induced acute lung injury[60]. In our study, we also found that EphA2 knockdown could effectively improve regorafenib-induced hepatotoxicity, which suggests that reducing the impact of EphA2 inhibition could also be a fascinating intervention strategy. EphA2 plays a negative role in regulating disease progression, which may be facilitated by the abnormal regulation of phosphorylation. The phosphorylation of EphA2 is widely involved in various life activities, especially in tumor cells. In normal cells, the relationship between EphA2 phosphorylation and cell survival has not been fully discussed. Here, we demonstrate that EphA2 Ser897 phosphorylation could maintain liver homeostasis by regulating the level of p53, while EphA2 Tyr588 phosphorylation rarely has an influence on p53 level regulation and hepatoxicity. Phosphorylation of EphA2 at Tyr588 is considered ligand-dependent activation and exhibits an inverse relationship with the phosphorylation of Ser897 upon mitotic entry[61]. GLPG1790, a small molecule targeting various Eph receptor kinases, could inhibit the phosphorylation of Ser897 and Tyr588 and completely block the EphA2 signaling pathway, exhibiting an antiglioma effect[62]. In our study, regorafenib inhibited EphA2 at both sites in hepatocytes. However, p-EphA2^Tyr588 was not involved in p53-induced apoptosis or regorafenib-

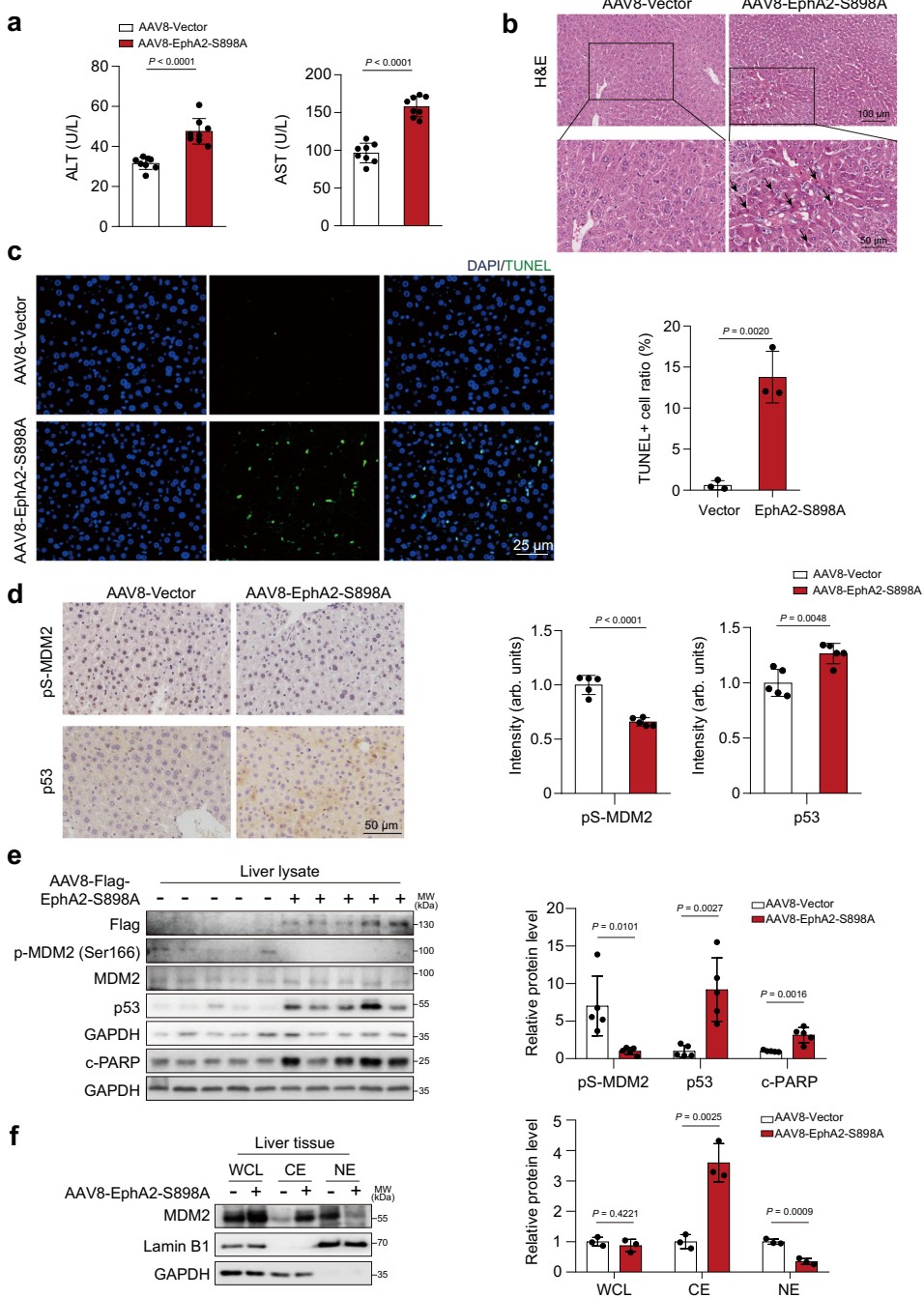

**Fig. 6 | Hepatocyte-specific overexpression of EphA2-S898A induced hepatocytes apoptosis by increasing p53 levels in vivo. a–f** C57BL/6J male mice were injected with AAV8-TBG-Vector or AAV8-TBG-EphA2-S898A through tail vein for 6 weeks (*n* = 8 per group). **a** The levels of serum ALT and AST were measured (*n* = 8 per group). **b** Representative images of H&E staining in liver tissues (*n* = 8 per group). Black arrowheads indicated nuclear shrinkage or structure disorder with vacuolization in specific region. For 100× magnification, scale bar: 100 μm; for 200× magnification, scale bar: 50 μm. **c** Representative images of TUNEL staining (green) in liver tissues. One area from three mice per group. Scale bar: 25 μm. **d** Representative images of immunohistochemistry for p53 and p-MDM2 (Ser166) staining in liver tissues. Scale bar: 50 μm. The staining was quantified by densitometric analysis. One area from five mice per group. **e** The expression levels of Flag, p-MDM2 (Ser166), MDM2, p53 and c-PARP of liver lysate were analyzed by western blot. *n* = 5 biologically independent samples per group. **f** The expression levels of MDM2 and Lamin B1 in whole cell, cytoplasm and nuclei of liver tissues were detected by western blot. *n* = 3 independent experiments. Data were expressed as mean ± SD. Unpaired two-sided Student's *t* test for **a**, **e**, and **f**. Source data are provided as a Source Data file. MW molecular weight, WCL whole cell lysate, CE cytoplasmic extraction, NE nuclear extraction.

induced hepatotoxicity. Whether p-EphA2$^{Tyr588}$ has other functions under regorafenib treatment remains to be further studied.

Based on its physiological function, p53 plays an irreplaceable role in the liver. The overexpression of p53 is involved in steatosis from nonalcoholic fatty liver disease to nonalcoholic steatohepatitis[63]. p53 functional regulation has been described at the level of transcription,

translation, degradation, structural alterations, and posttranslational modification[64]. The key mechanism of p53 regulation is mainly dependent on the control of protein stability[65]. MDM2, the primary suppressor of p53, was reported to inhibit p53 by regulating its stability, cellular localization, and transactivation. The signaling pathway or crucial regulatory factor-induced p53 is mainly mediated by the

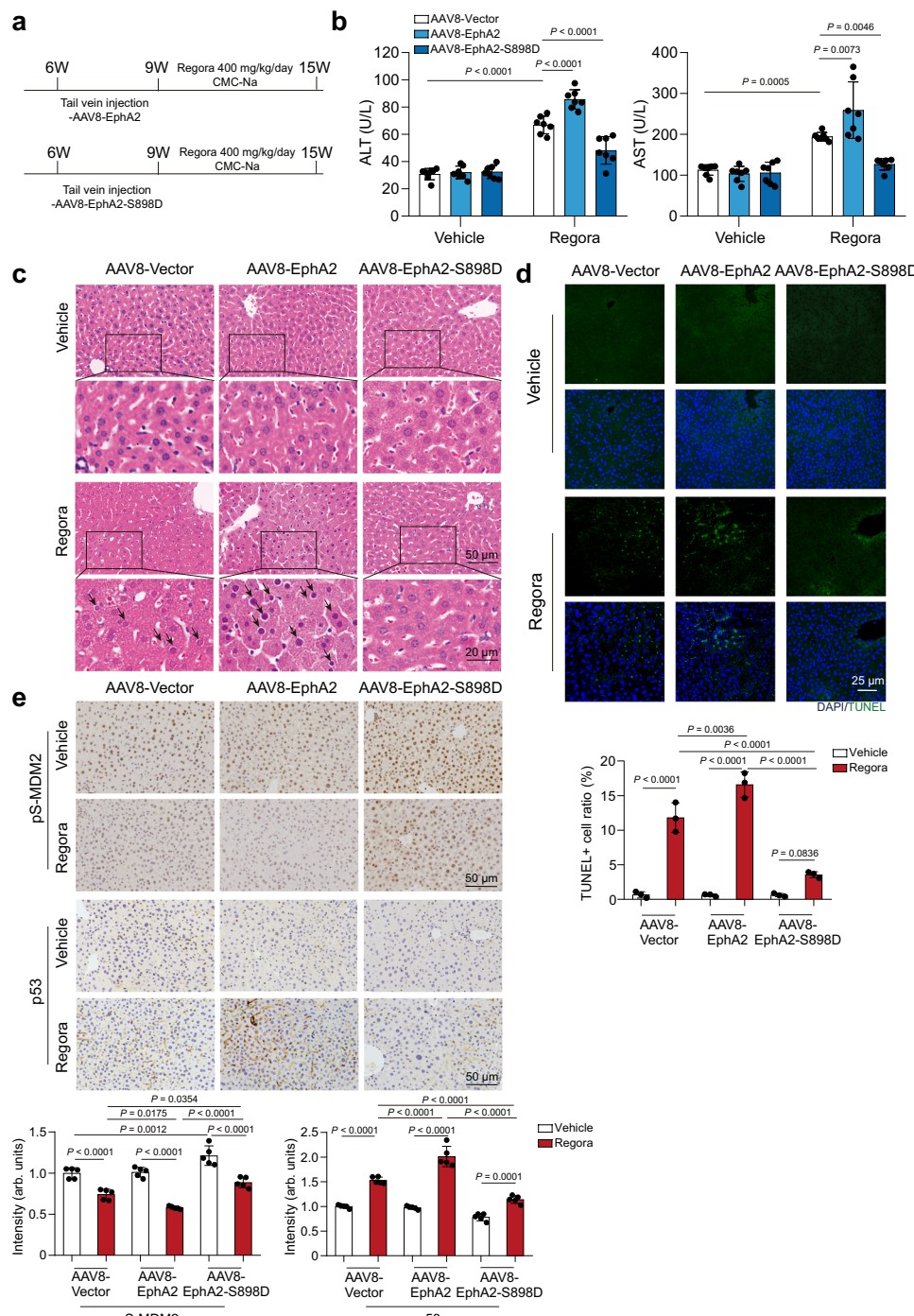

**Fig. 7 | Hepatocyte-specific overexpressing EphA2-S898D attenuated regorafenib-induced hepatotoxicity in vivo. a–f** AAV8-TBG-Vector, AAV8-TBG-EphA2 or AAV8-TBG-EphA2-S898D were injected into C57BL/6J male mice through tail vein. Three weeks later, mice were treated with 0.5% CMC-Na or 400 mg/kg/day regorafenib ($n$ = 7 per group). **a** Diagram of the construction of mouse model and treatment. **b** The levels of serum ALT and AST were analyzed ($n$ = 7 per group). **c** Representative images of H&E staining in liver tissues ($n$ = 7 per group). Black arrowheads indicated nuclear shrinkage or structure disorder with vacuolization in specific region. For 200× magnification, scale bar: 50 μm; for 500× magnification,

scale bar: 20 μm. **d** Representative images of TUNEL staining (green) in liver tissues. One area from three mice per group. Scale bar: 25 μm. **e** Representative images of immunohistochemistry for p-MDM2 (Ser166) staining and p53 staining in liver tissues. Scale bar: 50 μm. The immunohistochemistry for p-MDM2 (Ser166) and p53 staining was quantified by densitometric analysis. One area from five mice per group. Data were expressed as mean ± SD. One way ANOVA followed by Tukey post hoc test (**b**, **d** and **e**). Source data are provided as a Source Data file. Regora regorafenib.

posttranscriptional modification of MDM2, which is responsible for its function and localization. Phosphorylation of MDM2 at Ser359 and Tyr394 was reported to decrease the ability of MDM2 to bind p53 for degradation. Wip1 can catalyse the dephosphorylation of MDM2 at Ser359, resulting in MDM2 stability[66]. Hedgehog signaling induces the

phosphorylation of MDM2 at Ser166 and Ser168, which are the activating sites for MDM2[67]. Activated AKT and ERK can phosphorylate MDM2 at Ser166, which causes MDM2 to enter the nucleus and transport p53 to the cytoplasm for degradation. These phosphorylation sites of MDM2 may serve to protect against p53-induced apoptosis. In our study, we

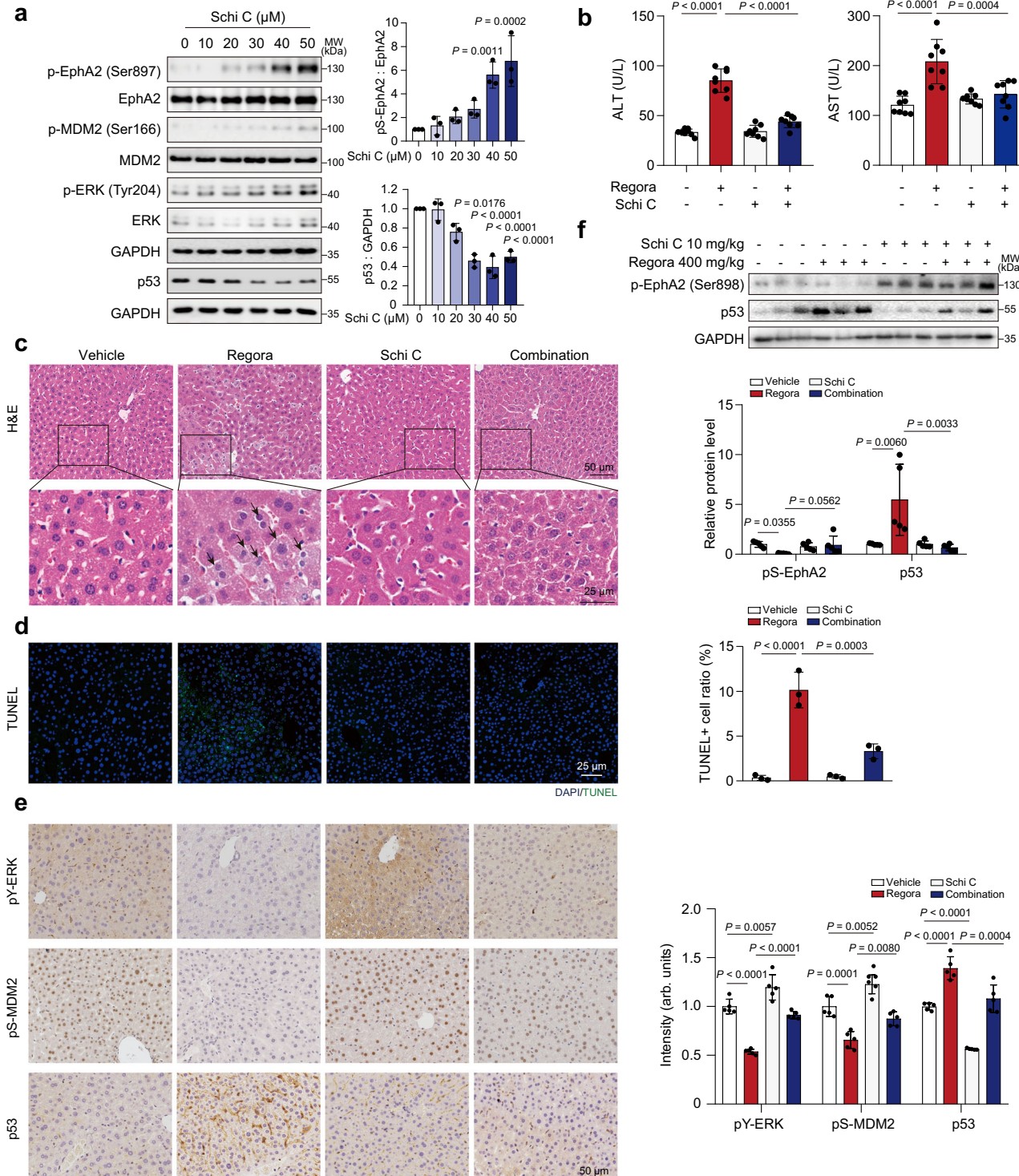

**Fig. 8 | Schisandrin C attenuates hepatocytes apoptosis by activating the phosphorylation of EphA2$^{Ser897}$ in vivo. a** HL-7702 cells were treated with 0, 10, 20, 30, 40 or 50 μM schisandrin C for 24 h. The expression levels of p-EphA2 (Ser897), EphA2, p-MDM2 (Ser166), MDM2, p-ERK (Tyr204), ERK and p53 were detected by western blot. $n = 3$ independent experiments. **b–f** C57BL/6J male mice were treated with 400 mg/kg/day regorafenib and/or 10 mg/kg/day schisandrin C for 6 weeks. **b** The levels of serum ALT and AST were analyzed ($n = 8$ per group). **c** Representative images of H&E staining in liver tissues ($n = 8$ per group). Black arrowheads indicated nuclear shrinkage or structure disorder with vacuolization in specific region. For 200× magnification, scale bar: 50 μm; for 500× magnification, scale bar: 20 μm. **d** Representative images of TUNEL staining (green) in liver tissues. One area from three mice per group. Scale bar: 25 μm. **e** Representative images of immunohistochemistry for p-ERK (Tyr204), p-MDM2 (Ser166) and p53 staining in liver tissues. Scale bar: 50 μm. The immunohistochemistry for p-ERK (Tyr204), p-MDM2 (Ser166) and p53 staining was quantified by densitometric analysis. One area from five mice per group. **f** The expression levels of p-EphA2 (Ser898) and p53 in liver tissues were analyzed by western blot. $n = 5$ independent samples examined over 2 independent experiments. Data were expressed as mean ± SD. One way ANOVA followed by Tukey post hoc test for **a**, **b**, **d**–**f**. Source data are provided as a Source Data file. Schi C Schisandrin C, Regora regorafenib, MW molecular weight.

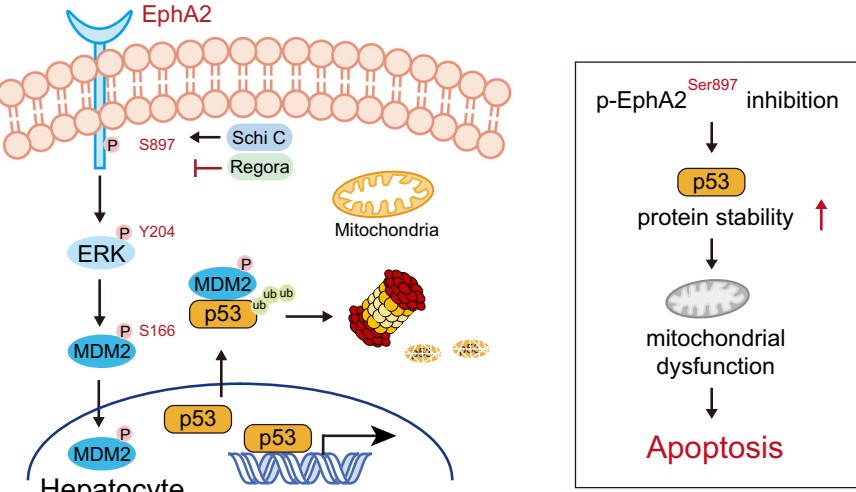

**Fig. 9 | Schematic diagram showing the mechanism of regorafenib-induced hepatotoxicity.** Regorafenib inhibits the phosphorylation of EphA2 at Ser897, which decreases the phosphorylation of ERK at Tyr204, thereafter contributing to remain MDM2 in the cytoplasm. p53 cannot be transported out of the nucleus by MDM2 and its abnormal accumulation eventually leads to hepatocyte mitochondria-dependent apoptosis.

demonstrated that the nuclear localization of MDM2 was significantly decreased after treatment with regorafenib and the S897A mutant plasmid as well as the reduced ubiquitination of p53. The phosphorylation inhibition of EphA2 at Ser897 was accompanied by decreased ERK phosphorylation, which mediated the inactivation of MDM2.

Schisandrin C is one of the representative lignans from the extracts of *Schisandra chinensis*, which is believed to be a hepatic tonic in China, Japan and Russia. Schisandrin C has been reported to have a protective role against hepatocyte necrosis, fatty degeneration, and inflammatory cell infiltration in hepatitis and is an intervention in CCl₄-induced hepatotoxicity[68,69]. Here, we discovered that schisandrin C could induce the phosphorylation of EphA2 at Ser897, recover the p53 level and rescue hepatocyte death and liver dysfunction caused by regorafenib. This finding enhances our understanding of the role of EphA2 Ser897 phosphorylation in regulating p53 levels and confirms the feasibility of schisandrin C as a candidate drug for the treatment against the hepatotoxicity of regorafenib. Schisandrin C and its analog led to the development of the potent derivative BICYCLOL, which is currently in clinical trials against HBV and acute DILI (NCT02944552, NCT05063500)[70]. To date, we have performed research on Wu wei zi granules (Chinese patented drug, its main component is Schisandrae Chinensis Fructus) to intervene in the hepatotoxicity of regorafenib in cooperation with the clinic. Patients who achieved grade 1-2 elevated ALT and AST were enrolled in this study. After administration of Wu wei zi granules at 10 g/tid for 2 weeks, the levels of ALT and AST in 15 patients were well-controlled without adjusting the dosage of regorafenib. However, whether Wu wei zi granules influence the therapeutic effect and hepatotoxicity of regorafenib over a long course merits further investigation.

Taken together, our research reveals a system of drug-induced hepatotoxicity where the nontherapeutic target of EphA2 is vital under regorafenib treatment. We further found that the phosphorylation of EphA2 Ser897 regulates p53 levels by removing ubiquitin through the ERK/MDM2 axis. Combining schisandrin C to recover the phosphorylation of EphA2 Ser897 may be a safe and effective strategy for cancer treatment.

## Methods
### Cell culture
The human primary hepatocytes were provided by BioreclamationIVT (F00995-P and M00995-P, Hicksville, New York, USA), and the clinical characteristics of the donors are shown in Supplementary Table 1.

Human primary hepatocytes were thawed in INVITROGROTM CP Medium (S03316, BioreclamationIVT, Hicksville, New York, USA) supplemented with 10% fetal bovine serum (16140071, Gibco, New York, USA) at 37 °C in a 5% CO₂ incubator for 24 h and then replaced with INVITROTM HI Medium (Z99009, Hicksville, New York, USA). HL-7702 (JNO-048) was purchased from GuangZhou Jennio Biotech Co.,Ltd. (Guangzhou, Guangdong, China), SW-480 (TCHu 86), HCT-116 (TCHu 99), Bel-7402 (TCHu 10), Hep-G2 (TCHu 72), and HEK-293T (GNHu 18) cells were originally obtained from the Cell Bank of China Science Academy. HL-7702, SW-480, HCT-116 and Bel-7402 cells were maintained in RPMI-1640 (21870076, Gibco, New York, USA), while HEK-293T and Hep-G2 cells were maintained in DMEM (10569010, Gibco, New York, USA) with 10% fetal bovine serum (16140071, Gibco, New York, USA), 100 U/mL penicillin and 100 µg/mL streptomycin (15140122, Gibco, New York, USA) in a humidified atmosphere with 5% CO₂. All cell lines were routinely tested to be negative for mycoplasma contamination.

### Animal experiments
All experimental procedures and methods in mice were approved by the Center for Drug Safety Evaluation and Research of Zhejiang University and performed according to the Institutional Animal Care and Use Committee (IACUC) protocol of Zhejiang University. 6 weeks old male mice or female mice (about 18–22 g) on C57BL/6J background were purchased from Shanghai Laboratory Animal Center (Shanghai, China). The mice were housed in pathogen-free conditions in individual metabolic cages under a 12-h/12-h light/dark cycle barrier facility with stable temperature (21–23 °C) and humidity (40–70%). General laboratory diet and water were freely available. Approximately 1 week was provided before starting experiments to allow the animals to adapt to the laboratory environment. Before sample collection, mice were fasted for 12 h. Mice were euthanized by cervical dislocation under isoflurane. In the study of regorafenib-induced hepatotoxicity, regorafenib was dissolved in 0.5% CMC-Na, and the mice were treated with 200 mg/kg/day regorafenib for 3 weeks or 400 mg/kg/day regorafenib for 3 or 6 weeks by gavage (*n* = 5–8 per group). To determine whether schisandrin C has therapeutic potential for regorafenib-induced liver injury, male mice were treated with 400 mg/kg/day regorafenib and/or 10 mg/kg/day schisandrin C for 6 weeks (*n* = 8 per group).

Specific knockdown of EphA2 in mouse hepatocytes is achieved by adeno-associated virus (AAV). A liver-specific adeno-associated

virus serotype 8 (AAV8), allowing for hepatocyte-targeted RNAi against EphA2 (AAV8-TBG-sh*EphA2*) was constructed and packaged by Vigene Biosciences (Rockville, MD, US). The AAV8-control ($3.87 \times 10^{13}$ vg/mL) and AAV8-sh*EphA2* ($4.13 \times 10^{13}$ vg/mL) were then injected into C57BL/6J male mice through the tail vein ($4 \times 10^{11}$ vg per mouse) ($n = 8$ per group).

Specific overexpression of EphA2 in mouse hepatocytes is also achieved by AAV. A liver-specific AAV8, allowing for hepatocyte to overexpress EphA2 (AAV8-TBG-EphA2), EphA2-S898A (AAV8-TBG-EphA2-S898A) or EphA2-S898D (AAV8-TBG-EphA2-S898D) was constructed and packaged by Shanghai GeneChem Co., Ltd. China (Shanghai, China). The AAV8-vectors ($1.02 \times 10^{13}$ vg/mL), AAV8-EphA2-S898A ($1.26 \times 10^{13}$ vg/mL) and AAV8-EphA2-S898D ($4.19 \times 10^{13}$ vg/mL) were then injected into C57BL/6J male mice through the tail vein ($4 \times 10^{11}$ vg per mouse) ($n$ = 7-8 per group).

## Materials

Regorafenib (Regora; R-0443565) was provided by Heowns (Tianjin, China). Chloroquine (CQ; C6628) and MG132 (M8699) were purchased from Sigma-Aldrich (Shanghai, China). Z-VAD-FMK (C1202) was purchased from Beyotime (Shanghai, China). ALW-II-41-27 (HY-18007), cycloheximide (CHX; HY-12320), TBHQ (HY-100489) and EFNA1 recombinant protein (HY-P72662) were provided by MedChemExpress (New Jersey, USA). Schisandrin C (Schi C; FY1671) was provided by Feiyubio (Nantong, Jiangsu, China). Ponatinib (Pona; T2372), dasatinib (Dasa; T1448), nilotinib (Nilo; T1524), bosutinib (Bosu; T0152) and crizotinib (Crizo; T1661) were purchased from Topscience (Shanghai, China). Antibodies used for western blot, immunofluorescence and immunohistochemistry analyses are showed in Supplementary Table 2. FDA-approved drugs and natural products for drug screening were purchased from Topscience (Shanghai, China) and Nantong Jingwei Biotechnology Co., Ltd (Nantong, Jiangsu, China), respectively. Related information was shown in Supplementary Table 3.

## Biochemistry assessment of liver injury in C57BL/6J mice

After blood was collected from mouse orbit, the blood samples were left for 2 h, centrifuged at $3400 \times g$ for 10 min, and the serum was collected for detection. Serum alanine aminotransferase (ALT) and aspartate aminotransferase (AST) levels were measured by automated chemical analyser (XN-1000V, Sysmex, Kobe, Japan).

## Protein extraction and western blot

Total protein was extracted from cells or liver tissues by using a RIPA Kit (P0013B, Beyotime, Shanghai, China). Protein lysates (30–50 μg per sample) were loaded and run on 8%, 10% or 12% SDS-polyacrylamide gels, transferred to a PVDF membrane (IPVH00010, Merck Millipore, MA, USA), incubated overnight with primary antibody at 4 °C. Next, membranes were washed three times by using PBS with 0.1% Tween-20 (T-PBS) for 15, 5 and 5 min, and incubated with secondary antibodies for 1 h at room temperature. After being washed three times with T-PBS, membranes were incubated with Western Lightning Plus-ECL reagent (NEL105001EA, PerkinElmer, Waltham, MA, USA) according to the manufacturer's instructions. Finally, membranes were exposed using Amersham Imager 600 (General Electric Company, Boston, MA, USA). The following antibodies were used: anti-GAPDH (db106, 1:10000, Diagbio, Hangzhou, Zhejiang, China), anti-p53 (sc-126, 1:1000, Santa Cruz Biotechnology, Texas, USA), anti-EphA2 (#6997, 1:1000, Cell Signaling Technology, Boston, MA, USA), anti-phospho-EphA2 (Ser897) (#6347, 1:1000, Cell Signaling Technology, Boston, MA, USA), anti-phospho-EphA2 (Tyr588) (#12677, 1:1000, Cell Signaling Technology, Boston, MA, USA), anti-MDM2 (RT1382, 1:1000, Huabio, Hangzhou, Zhejiang, China), anti-MDM2 (A13327, 1:1000, ABclonal, Wuhan, Hubei, China), anti-phospho-MDM2 (Ser166) (#3521, 1:1000, Cell Signaling Technology, Boston, MA, USA), anti-c-PARP

(ET1608-10, 1:1000, Huabio, Hangzhou, Zhejiang, China), anti-HA tag (db2603, 1:1000, Diagbio, Hangzhou, Zhejiang, China), anti-c-PARP (ab32064, 1:1000, Abcam, Cambridge, UK), anti-FLAG tag (db7002, 1:1000, Diagbio, Hangzhou, Zhejiang, China), anti-Lamin B1 (R1508-1, 1:1000, Huabio, Hangzhou, Zhejiang, China), anti-ERK (sc-135900, 1:1000, Santa Cruz Biotechnology, Texas, USA), anti-phospho-ERK (Tyr204) (sc-7383, 1:1000, Santa Cruz Biotechnology, Texas, USA), anti-LC3 (#4108, 1:1000, Cell Signaling Technology, Boston, MA, USA), anti-phospho-Akt (Ser473) (#4060, 1:1000, Cell Signaling Technology, Boston, MA, USA), anti-EFNA1 (A9132, 1:1000, ABclonal, Wuhan, Hubei, China). HRP-labeled secondary antibodies (GAR007, 1:1000, GAM007, RAG007) were purchased from LiankeBio (Hangzhou, Zhejiang, China). Image J (version 1.8.0) was used for densitometry analysis.

## SRB (Sulforhodamine B) colorimetry detects cell proliferation

Cell survival rate was assessed using sulforhodamine B (SRB; S1402, Sigma-Aldrich, Shanghai, China) colorimetric assay. In detail, SRB (4 mg/mL) was added to 96-well plates fixed with 10% trichloroacetic acid (T104257, Aladdin, Shanghai, China) after various treatments and incubated for 30 min. The liquid was discarded in a 96-well plate and washed more than 5 times with 1% acetic acid (1000218, Sinopharm, Shanghai, China). The bound SRB dye was dissolved in 10 mM unbuffered Tris lye (1115KG001, Biofroxx, Shanghai, China) and oscillated for 5 min until completely dissolved. The absorbance at 515 nm was measured using a multiscan spectrum (Thermo Fisher Scientific, Marietta, Ohio, USA) until the absorbance values remained unchanged. Assays were performed in 3 independent experiments.

## Flow cytometry analysis of Annexin V-PI staining

The apoptotic ratio was measured with a Pharmingen™ FITC Annexin V Apoptosis Detection Kit I (556547, BD Biosciences, New Jersey, USA). Procedures were performed according to the instructions. Briefly, cells were treated for the indicated time, harvested and washed with PBS for binding and Annexin V-PI staining. For each sample, $1 \times 10^4$ cells were acquired and analyzed by BD FACSCalibur™ Flow Cytometer (BD Biosciences, New Jersey, USA). Annexin V−PI−, Annexin V−PI+, Annexin V+PI−, Annexin V+PI+ staining represents viable cells, necrotic cells, early apoptotic cells, and late apoptosis cells being stained, respectively. The gate strategies were shown in Supplementary Fig. 23 recorded by BD CellQuest Pro software (version 5.1).

## Flow cytometry analysis of JC-1 staining

Mitochondrial membrane potential (hereafter referred to as MMP) was detected by JC-1 staining assays. After treatment with regorafenib for the indicated time, cells were collected by trypsinization and stained with JC-1 (5 μM; T4069, Sigma-Aldrich, Shanghai, China) for 30 min at 37 °C avoiding light. MMP was analyzed by BD FACSCalibur™ Flow Cytometer (BD Biosciences, New Jersey, USA). For each sample, $1 \times 10^4$ cells were measured by FACS-Calibur cytometer (BD Biosciences, New Jersey, USA). The gate strategies were shown in Supplementary Fig. 23 recorded by BD CellQuest Pro software (version 5.1).

## TUNEL staining

The sample was fixed in formalin (F8775, Sigma-Aldrich, Shanghai, China) and cut into 3 μm slice after paraffin embedding. Tissue sections were pre-treated proteinase K working solution (20 μg/mL in 10 mM Tris/HCl, pH 7.4-8, ST532, Beyotime, Shanghai, China) for 15 min at room temperature after the tissue sections were dewaxed and rehydrated. PBS rinse the slides 3 times. Add 50 μL of TUNEL reaction mixture (C1088, Beyotime, Shanghai, China) to the sample and incubate for 60 min in a dark and humidified chamber at 37 °C. After three times PBS rinse, DAPI (D212, 1:5000, Dojindo, Beijing, China) was used to stain the nuclei. The immunofluorescent images were captured under a fluorescence microscope (Olympus, Tokyo,

Japan). The image was then quantified and analyzed by Image J (version 1.8.0).

## Hematoxylin-eosin staining

After dewaxed and rehydrated, the liver tissue paraffin sections were stained in hematoxylin (C0105, Beyotime, Shanghai, China) for 8 min and rinsed with tap water for 5 min. Next, the slices were stained in eosin (C0105, Beyotime, Shanghai, China) for 30 s. Finally, the sections were dehydrated and fixed with neutral resin to observe the morphology of the liver. The histological images were observed and captured under Aperio AT2 (Leica Biosystems, Heidelberg, Germany).

## Immunohistochemistry staining

After dewaxing, rehydration and antigen retrieval, liver tissue paraffin sections were pre-treated with 3% $H_2O_2$ (PV-6001, ZSGB-BIO, Beijing, China) at room temperature for 10 min and blocked with 5% goat serum (16210064, Gibco, New York, USA) for 30 min. The p53, phospho-MDM2 (Ser166) or phospho-ERK (Tyr204) expression profile in mice livers was determined by incubating the sections with blocking solution containing anti-p53 (TA502870, 1:100, Origene, MD, USA), anti-phospho-MDM2 (Ser166) (#3521, 1:150, Cell Signaling Technology, Boston, MA, USA) or anti-phospho-ERK (Tyr204) (sc-7383, 1:100, Santa Cruz Biotechnology, Texas, USA) at 4 °C overnights. The primary antibody was recognized by the horseradish peroxidase (HRP) conjugated secondary antibody (PV-6001, PV-6002, ZSGB-BIO, Beijing, China) for 60 min at room temperature. Finally, the slides were incubated with peroxidase substrate DAB kit (ZLI-9017, ZSGB-BIO, Beijing, China) and nuclei were stained in hematoxylin (C0107, Beyotime, Shanghai, China) for 3 s. The images were observed and captured under a light microscope (Olympus, Japan). Image J (version 1.8.0) was used to quantify positive stained regions. Three images were collected per sample, exported and saved as the original IHC images in.tiff file format. Image J (version 1.8.0) was used to set the minimum and maximum thresholds to zero, and arbitrarily determine a maximum threshold, thereby removing the background signal without removing the true DAB signal. Once selected, the maximum threshold setting for all IHC images was equal. The image was then quantified and analyzed by "area".

## Immunofluorescence

Cells grown on cell culture chamber are washed with PBS and then fixed with 4% paraformaldehyde (P6148, Sigma-Aldrich, Shanghai, China) for 15 min at room temperature. The cells were then permeabilized with 0.3% Triton X-100 (1139ML100, Biofroxx, Shanghai, China) in PBS for 10 min, blocked with 4% bovine serum albumin (BSA; B2064, Sigma-Aldrich, Shanghai, China) in Tris-buffered saline (20 mM Tris-HCl, 500 mM NaCl, pH 7.4) for 0.5 h and incubated with primary antibodies at 4 °C overnights. After washing with PBS, the cells were incubated with Alexa Fluor 488-, Alexa Fluor 568- or Alexa Fluor 647-conjugated secondary antibodies (Thermo Fisher Scientific, A21202, A10042, A31573, 1:200) for 1 h at room temperature, stained with DAPI (D212, 1:5000, Dojindo, Beijing, China) for 5 min and mounted for fluorescence microscopy. For immunofluorescence observation of liver tissues, frozen liver sections were fixed with 4% paraformaldehyde for 5 min and then permeabilized with 0.5% Triton X-100 for 5 min. Sections were washed and blocked with 4% bovine serum albumin (BSA; B2064, Sigma-Aldrich, Shanghai, China) in Tris-buffered saline (20 mM Tris-HCl, 500 mM NaCl, pH 7.4) for 0.5 h and incubated with primary antibodies at 4 °C overnights and secondary antibodies for 1 h at room temperature. Finally, sections were stained with DAPI (D212, 1:5000, Dojindo, Beijing, China) for 5 min and mounted on a fluorescence microscope. The following primary antibodies were used: anti-p53 (sc-126, 1:100, Santa Cruz Biotechnology, Texas, USA), anti-EphA2 (#6997, 1:100, Cell Signaling Technology, Boston, MA, USA), anti-phospho-EphA2 (Ser897) (#6347, 1:100, Cell Signaling

Technology, Boston, MA, USA), anti-MDM2 (AF0208, 1:200, Affinity Biosciences LTD, Melbourne, Florida, USA), anti-phospho-ERK (Tyr204) (sc-7383, 1:200, Santa Cruz Biotechnology, Texas, USA).

## Immunoprecipitation (IP)

Protein lysates were prepared in 50 mM Tris-HCl, pH 7.5, 150 mM NaCl, 1 mM EDTA and 1% NP40 with protease inhibitor cocktail (#5871, Cell Signaling Technology, Boston, MA, USA) and were used for an overnight immunoprecipitation (500 μg per sample) at 4 °C with 10 μL of anti-Flag IP resin (L00425, GenScript, Nanjing, Jiangsu, China). Then the resin was washed with NETN buffer (0.5% NP-40, 1 mM EDTA, 100 mM NaCl and 20 mM Tris, pH 8.0) for 5 times before detection. The results of immunoprecipitation were analyzed using western blot assays.

## Nuclear/cytoplasmic fractionation

Perform nuclear/cytoplasmic isolation according to Nuclear/Cytosol Fractionation Kit (ab289882, Abcam, Cambridge, England). The liver tissues of animals should be ground before being lysed. HL-7702 cells were harvested by centrifugation and lysed into 100 μL of fractionation buffer per $5 \times 10^6$ cells. After centrifugation for 5 min on ice, the supernatant was the cytosolic part. The nucleus in the precipitate were added with 30 μL RIPA lysis buffer (P0013B, Beyotime, Shanghai, China). The nucleus and the cytosol were centrifuged at $16,000 \times g$ for 30 min at 4 °C, respectively, then the supernatant was taken, and the corresponding volume of 2.5× loading was added. The changes of protein levels were detected by loading samples in western blot system.

## Quantitative real time polymerase chain reaction (qRT-PCR)

Total RNA was extracted according to the manufacturer's protocol for TRIzol Reagent (15596026, Invitrogen, Carlsbad, CA, USA). Equal amount of RNA was reverted to cDNA by using a cDNA reverse transcription kit (AT311-02, Transgene Biotech, Beijing, China). qRT-PCR was performed by using iTaq Universal SYBR Green Supermix (1725125, Bio-Rad, CA, USA) in a CFX96TM Real-Time System (Bio-Rad, CA, USA). The samples underwent two-step amplification with an initial step at 95 °C (3 s) and 60 °C (31 s) for 39 cycles. The melting curve was analyzed. Fold changes in the expression of each gene were calculated by the comparative threshold cycle method. Each sample was set in duplicated wells. Each experiment was performed in triplicate independently. The sequences of primers purchased from Biosune (Shanghai, China) are shown in Supplementary Table 4.

## Transfection of siRNA oligonucleotides

siRNA oligonucleotides were transfected using oligofectamine™ transfection reagent (12252011, Invitrogen, Carlsbad, CA, USA) at a final concentration of 40 nM. The medium in the 6-well plate was discarded, and the prepared transfection system solution was added to 6 wells containing opti-MEM™ (31985070, Gibco, New York, USA). After 6 h of transfection, the medium was replaced with complete medium, and the corresponding drug stimulation was added according to the experimental design. siRNA and a negative control siRNA (NC) were synthesized by GenePharma (Shanghai, China). The siRNA sequences are shown in Supplementary Table 5.

## Plasmid construction and transfection

The overexpression plasmid pCMV6-Entry-EphA2-Myc-DDK was purchased from OriGene (RC205725, Origene, MD, USA). The cDNA of EphA2 was amplified and subcloned to the vector with pcDNA3.0-HA. pcDNA3.0-EphA2-HA was generated by restriction digestion of the plasmids with HindIII (R3104V, NEB, MA, USA) plus XhoI (R0146V, NEB, MA, USA), followed by ligations. The cDNA of EFNA1 was amplified from the reversed RNA and subcloned to the vector with pcDNA3.0-

Flag. pcDNA3.0-EFNA1-Flag was generated by restriction digestion of the plasmids with BamHI (R3136V, NEB, MA, USA) plus XhoI (R0146V, NEB, MA, USA), followed by ligations. One-site mutant plasmids of pcDNA3.0-EphA2-S897A-HA, pcDNA3.0-EphA2-S897D-HA, pcDNA3.0-EphA2-Y588A-HA and pcDNA3.0-EphA2-K646M-HA were generated by using Hieff Mut™ Site-Directed Mutagenesis Kit (11003ES10, Yeasen, Shanghai, China). Cells were transfected using Lipofectamine 2000 (11668019, Invitrogen, Carlsbad, CA, USA) according to the instructions. Briefly, cells were seeded and grown to ~70% confluence. Related plasmid was transfected into the cells with transfection reagent for 4–6 h and then replaced with fresh culture medium.

## ATP detection

ATP test kit (S0027, Beyotime Biotechnology, Shanghai, China) was used to detect the ATP of the samples. The culture solution was removed and lysate was added in the proportions specified for cell lysis. After cracking and centrifugation, supernatant was taken for subsequent determination. ATP test solution was prepared according to the proportion specified in the instructions, and added into the test hole, and placed at room temperature for 3–5 min. 20 μL samples or standards are added to test holes or tubes for rapid mixing, measuring relative light unit (RLU) with luminometer at least 2 s intervals. The ATP standard solution was diluted to an appropriate concentration gradient and the standard curve was prepared according to the instructions. Briefly, the RLU was measured by luminometer using a spark® multifunctional microplate detector (Paramit, CA, USA) until the absorbance values remained unchanged.

## MitoTracker staining

Mitochondria content in HL-7702 cells was evaluated by selectively loading mitochondria with deep red fluorescent dye MitoTracker (M22426, Invitrogen, Carlsbad, CA). The cells were grown on the lid of a petri dish containing the appropriate medium. When the cells reached the desired degree of fusion, the medium was removed from the dish and a preheated (37 °C) staining solution containing 100 nM MitoTracker probe was added. They were cultured for 25 min under growth conditions suitable for hepatocytes and photographed using a fluorescence microscope (Olympus, Tokyo, Japan).

## Statistical analysis

Data are represented as averaged (mean) ± standard deviation (SD). Student's $t$ test (unpaired, two side) was used to analyze significant differences between two groups, and one-way ANOVA was used to analyze significant differences between three or more groups. $P < 0.05$ was considered statistically significant. Data analysis was performed using Microsoft Excel version 1808, Image J (version 1.8.0), and GraphPad Prism Software Version 8.0 (GraphPad Prism, San Diego, California, USA). All the statistical details of the experiments can be found in the figure legends, including the exact number of cells or mice. No data were excluded from any of the experiments.

## Reporting summary

Further information on research design is available in the Nature Portfolio Reporting Summary linked to this article.

## Data availability

The authors declare that the main data supporting the findings of this study are available in this Article, its Supplementary Information and Source Data. Extra data for the individual measurements are available on request. The source data underlying Fig. 1b-e, g-i, 2a-f, h-l, 3b-e, g-j, 4b-d, f-h, 5a-d, f-h, 6a, c-f, 7b, d-e, 8a-b, d-f as well as Supplementary Fig. 1a, 1d, 2a-e, 3a-b, 3d, 4a-b, 5a-c, 7a, 8a-e, 9, 11a-c, 13a-d, 14b, 16a-b, 17a-d, 18a-b, 18d, 19b-c, 20, 21 are provided as a Source Data file. Source data are provided with this paper.

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

## Acknowledgements
We acknowledge financial support by Nature Science Foundation of Zhejiang Province (No. LQ22H310002 to H.Y.), National Natural Science Foundation of China (No. 82173893 to P.L.) and Key Laboratory of Clinical Cancer Pharmacology and Toxicology Research of Zhejiang Province (2020E10021 to H.Y.).

## Author contributions
B.Y., Q.H. and P.L. designed the research studies. H.Y., W.W., Y.H., J.L., J.X., X.C., Z.X. and X.Y. performed the experiments, acquired, and analyzed the data. H.Y. and P.L. contributed to the funding acquisition and project administration. H.Y., W.W., J.L. and P.L. wrote the manuscript. All the authors approved the final version of the manuscript.

## Competing interests
The authors declare no competing interests.
