## [Peer Review File · Nature Communications]

Regorafenib inhibits EphA2 phosphorylation damages the liver via ERK/MDM2/p53 axisEditorial Note: Parts of this Peer Review File have been redacted as indicated to remove third-party material where no permission to publish could be obtained.

REVIEWER COMMENTS

Reviewer #1 (Remarks to the Author):

Thank you for the opportunity to review the manuscript from Yan et al. which demonstrates that the inhibition of EphA2 Ser897 phosphorylation is a central factor of regorafenib-induced hepatotoxicity, and chemical activation on EphA2 Ser897 represents a potential therapeutic strategy to prevent regorafenib-induced hepatotoxicity. Overall the manuscript is well-present and methodology adequate, involving experiments in vitro and in vivo. Despite the overall good quality of this work, I have some important (and other minor) questions, which I would like to describe.

- 1) The animal model used should be justified. Why only male were used? Despite obvious hormone-related differences, toxicological assessment in both sexes is more translated to what is observed in humans.
- 2) One major problem of this study is the very high dose of regorafenib (200 and 400 mg/kg/d). What about body weight, daily food intake and daily water intake? The authors have to indicate these data in order to evaluate the effects of this high dose on the health status of mice. How can they justify this dose since there is no correlation with clinical situation?
- 3) Line 138-140: Please rephrase this sentence.
- 4) The authors found a dissipation of MMP. How can they explain this result? Is it due to decreased mitochondrial respiration, decreased mitochondrial mass? The authors should measure one of these parameter.
- 5) It is not clear whether the effect on mitochondria is an on or off-target effect? What is exactly the link to EphA2. Please discuss.
- 6) The authors found that regorafenib induces apoptosis. This result is not so novel. Please provide the relevant references.

Reviewer #2 (Remarks to the Author):

This manuscript by Yan and colleagues assesses the mechanism of regorafenib-induced liver toxicity. Briefly, they provide evidence that regorafenib induces hepatocyte death and mitochondrial dysfunction, induce p53 protein stability, and suppress phosphorylation of EphA2, ERK1/2, and MDM2. The authors utilized hepatocyte cell lines, human primary hepatocytes, and immunocompetent mice to assess the toxicity of regorafenib. This is an interesting manuscript that could have significant implications for the field. The work presented is sound and rigorous, but most evidence is correlative and key studies are lacking to fully elucidate the mechanism. Additionally, the conclusions drawn from some key experiments exhibit some flaws. These significant concerns and others are outlined below and should be addressed.

1. In Figure 2e-f, it is suggested that p53 is driving cell death in regorafenib-treated hepatocytes. Knockdown of p53 did not rescue the apoptosis induced by regorafenib, only modestly decreasing cell

death detected by PI/Annexin-V staining. The same is true for survival (Figure 2c). However, a p53 knockdown resulted in a more substantial recovery of mitochondrial membrane potential (JC-1 staining) and PARP cleavage. Likewise, EphA2 knockdown decreases PARP-cleavage and p53 levels (Figure 3c) but had only a modest impact on PI/Annexin-V cell death (Supplementary Figure 5d) or even cell survival (Figure 3b). Regorafenib was found to induce cell death in colorectal cancer cells via PUMA that was not dependent on p53 mutational status (Chen D, et al. Clin Cancer Res. 20(13):3472-3484. 2015.) Do EphA2 and p53 rescue mitochondrial function and not cell death?

2. If EphA2 S897 phosphorylation was the key driver in p53 upregulation and cell death, then expression of EphA2-S897A should lead to increased p53 protein even in the absence of regorafenib (Supplementary Figure 9). However, regorafenib was still required to increase p53 protein. Is EphA2 kinase activity required to stabilize p53?

3. Ser897 phosphorylation of EphA2 is indicative of ligand-independent activation, while Tyr588 phosphorylation is indicative of ligand-dependent activation. How does ephrin-A1 impact the proposed signaling pathway? Does regorafenib-induced toxicity change ephrin-A1 expression in the liver?

4. The data are not sufficient to support the mechanism of ERK1/2 on MDM2 phosphorylation. Regorafenib decreases ERK1/2 phosphorylation, but this may be due to inhibition EphA2 or other receptor tyrosine kinases. Data shown in Supplementary Figure 15 is proposed to show that ERK1/2 phosphorylation is associated with EphA2 S897 phosphorylation status. However, the differences are quite weak and based only on a single cell with no quantitation shown. Additionally, ERK has also been shown to regulate EphA2 S897 phosphorylation (Hamaoka Y, et al. Cell Signal. 28(8):937-945. 2016.), indicating that ERK may be upstream of EphA2. Please address.

5. In several blots, regorafenib appears to modestly reduce the protein levels of EphA2 (for example, Figure 3c, Figure 4b). Are expression levels of EphA2 reduced in regorafenib-treated cells? EphA2 phosphorylation should be normalized to total EphA2 and not GAPDH.

6. On lines 320-321, the authors indicated that regorafenib did not impact the distribution of EphA2, but this claim is not substantiated by any evidence. In fact, in Supplementary Figure 7b it appears that EphA2 may be more localized to the plasma membrane or at cell-to-cell interactions in regorafenib-treated hepatocytes. Please address.

7. In Figure 6, MDM2 is significantly lower in the nuclear fraction of AAV-EphA2-898A livers but is unchanged in the WCL. This discrepancy does not appear to be associated with an increase in the cytoplasmic fraction, which is suggested in Supplementary Figure 15. This is also true in regorafenib-treated cells and livers shown in Figure 5b. If not the cytoplasm or degradation, where is MDM2 going?

8. The knockdown of EphA2 in Figure 3 is quite poor, making it surprising to see complete recovery of p53 and c-PARP. In addition, additional EphA2 siRNAs should be used to increase confidence that these impacts on c-PARP and p53 are not due to off-target effects.

9. The authors suggest liver morphology changes, including nuclear size and immune infiltration, occur in mice treated with regorafenib (Figure 1a), infected with AAV (Figures 6b, 7c), and treated with regorafenib +/- Schisandrin C (Figure 8c). These observations, particularly those made on AAV models, are not obvious differences compared to control livers. Were these observations made in collaboration with a pathologist?

10. On lines 385-386, it was indicated that a drug screen was used to identify Schisandrin C as a compound that increases EphA2 phosphorylation. Please show the data from the screen.

11. Does Schisandrin C change phosphorylation of other EphA2 sites? If the EphA2 kinase domain is mutated, does this abrogate the effects of Schisandrin C?

12. Can the authors address why Schisandrin C rescues regorafenib-induced liver toxicity (Supplementary Figure 16) but has no impact on regorafenib efficacy on hepatocellular carcinoma (Supplementary Figure 19)?

13. It is unclear why no statistical comparisons are not shown between samples treated with vehicle or regorafenib in siRNA/EphA2 mutant experiments. For example, in Figure 2e apoptosis is still quite elevated in p53-knockdown cells treated with regorafenib compared to those treated with vehicle. This occurs throughout the manuscript and can have a significant impact on data interpretation. Please include these statistical comparisons.

14. Several of the blots and images shown do not appear to be representative of the corresponding quantitated data. In Supplementary Figure 2f, the bar graph shows the c-PARP in Regorafenib + Z-VAD-FMK is still considerably higher than the Z-VAD-FMK alone lane, but there is no difference in the blot. In the EphA2 knockdown lysates shown in Figure 3c, cleaved PARP and p53 appears to be higher in the regorafenib lane than the untreated lane, but the combined data shows no difference. Showing values below the lane of each representative blots and individual data points on bar graphs for all plots may help.

Minor concerns:

15. Some of the fonts and scales, particularly on the flow cytometry plots, are impossible to read. Percentages for define populations should be indicated for representative data.

16. In some JC-1 staining experiments, it is unclear why the gates are drawn as they are as there is not clear indications of separate populations existing. This is particularly true in Figure 1d, Figure 2f, Supplementary Figure 5e. Do the authors have supporting controls for these experiments?

17. No quantified data is shown for the TUNEL images shown in Figure 3e.

Reviewer #3 (Remarks to the Author):

Comments:

Hepatotoxicity of regorafenib is one of the most noteworthy concerns for patients, however the mechanism is still unclear up to now. In this study, authors identified the inhibition of EphA2 Ser897 phosphorylation (human) as a key cause of regorafenib-induced hepatotoxicity, and chemical activation on EphA2 Ser897 represents a potential therapeutic strategy to prevent regorafenib-induced hepatotoxicity. This is a very interesting story and very significant for clinical HCC treatment with regorafenib.

The opinion of manuscript is very interesting, however, the data is too rough and not enough critical result is for supporting. The antibodies were not explained using for mice or human, or both in the method. For example, the authors said the anti-c-PARP antibody information ((ET1608-10, Huabio, Hangzhou, Zhejiang, China). However, the results showed that 25Kd cleaved band was detected in the mouse tissue (Fig 1f, no molecular weight information, we can find it in the primary data) and the band between 90kd to 100kd was detected in human cells and tissue. I think these results from different antibodies but the authors failed to show the antibodies information. All WB results are no molecular weight information. In addition, the EphA2 Ser898 or Ser897 phosphorylation in the

abstract is also very confusing to reader, because the authors didn't say the EphA2 Ser898 from mouse EphA2 gene and Ser897 from human EphA2 gene.

Point-to-point response:

Reviewer #1 (Remarks to the Author):

Thank you for the opportunity to review the manuscript from Yan et al. which demonstrates that the inhibition of EphA2 Ser897 phosphorylation is a central factor of regorafenib-induced hepatotoxicity, and chemical activation on EphA2 Ser897 represents a potential therapeutic strategy to prevent regorafenib-induced hepatotoxicity. Overall the manuscript is well-present and methodology adequate, involving experiments in vitro and in vivo. Despite the overall good quality of this work, I have some important (and other minor) questions, which I would like to describe.

1. The animal model used should be justified. Why only male was used? Despite obvious hormone-related differences, toxicological assessment in both sexes is more translated to what is observed in humans.

Response: Thanks for your kind remind. In the pivotal CORRECT phase III trial, a difference in toxicity between males and females has not been detected in a safety subgroup analysis (*Oncologist. 2019 Feb;24(2):185-192.*). According to a safety analysis with a relatively small sample size, the authors found that sex was the only factor associated with the development of grade ≥ 3 treatment-related adverse events (71% in females vs. 53% in males, $P = 0.035$). Though only rash, anorexia, diarrhea, hypertension, HFSR and fatigue were considered, little is known about the sex on regorafenib-induced hepatotoxicity (*Clin Colorectal Cancer. 2021 Dec;20(4):326-333.*). In general, there is not much difference between the data from the male or female mice, however, different genders have obvious differences in sensitivity to certain drugs and stimuli. Thus, we treated male and female C57BL/6J mice with 400 mg/kg regorafenib for 6 weeks and serum were collected every 3 weeks for biochemical examination. The results were shown in newly released Fig. S1b-d. The liver of both male and female mice showed similar changes in appearance after regorafenib treatment (Fig.S1b). Severe nuclear shrinkage could be observed in both sexes by HE staining (Fig.S1c). The level of ALT and AST slightly increased at the third week and

have changed greatly at sixth week. However, there is no significance between male and female (Fig. S1d).

Based on the label of regorafenib, age, sex, race, and weight were considered as no clinically meaningful effect on the pharmacokinetics of regorafenib. And from the results of single dose of regorafenib on male and female mice, we chose male mice for toxicological assessment in the rest of the animal experiments.

The label of regorafenib on FDA website:

https://www.accessdata.fda.gov/drugsatfda_docs/label/2020/203085Orig1s014lbl.pdf

Fig.S1b

Fig.S1c

Fig.S1d

Newly released figures

2. One major problem of this study is the very high dose of regorafenib (200 and 400 mg/kg/d). What about body weight, daily food intake and daily water intake? The authors have to indicate these data in order to evaluate the effects of this high dose on the health status of mice. How can they justify this dose since there is no correlation with clinical situation?

Response: Thanks for your advice. For the newly-added animal experiment, we detect the body weight every day, and the food, water intake every week. Form the chart, we

could see that regorafenib have an impact on body weight, food intake and water intake at 400 mg/kg/d when comparing with the vehicle group (Fig.R1). Regorafenib would slightly cause anorexia which was reflected on the food intake. The increment of water intake and food intake was abolished under regorafenib treatment, however, regorafenib wouldn't induce substantial reduction in body weight (loss of body weight $\geq 15\%$), food intake ($< 50\%$ of normal) and water intake. The weight increased steadily both sexes among 6 weeks. Thus, regorafenib would not have an obvious impact on the health status with 400 mg/kg/day.

In clinic, regorafenib could induce the increment of ALT and AST which suggests the damage of hepatocytes. The dose of regorafenib used in clinic is 160 mg/d and equivalent dose is about 24 mg/kg/d in mice. In the study of toxicology, we would use 10-20-fold clinical equivalent dose to test the long-term toxicity and increase the plasma drug exposure. We first test two dosage of regorafenib (about 8-fold and 16-fold) and found that only 400 mg/kg could greatly increase the level of ALT and AST, hepatocyte injury and structure disorder, which was consistent with the features of regorafenib-induced hepatotoxicity. Among the experiments *in vitro*, we used one-fold C_{max} of regorafenib (8 μM) to study the potential mechanism which is relevant to the clinical situation. Based on the findings, we carried out research of Wu wei zi granules (Chinese patent drug, main component is Schisandrae Chinensis Fructus) to intervene the hepatotoxicity of regorafenib in cooperation with the clinic (Hangzhou Cancer Hospital, Ethics number: HZCH2023-02/01). Patients who achieve grade 1-2 of elevated ALT and AST were enrolled in this study. After administration with Wu wei zi granules 10 g/tid for 2 weeks, the levels of ALT and AST of 15 patients were well controlled without adjusting the dosage of regorafenib. The outcome indicated our findings were relevant to the clinical situation.

Fig. R1

3. Line 138-140: Please rephrase this sentence.

Response: Thanks for pointing this out. We revised this sentence and made it more suitable for describing the related data. The rephrased sentence was as follows: Z-VAD-FMK, a pan-caspase inhibitor, could effectively protect hepatocytes against regorafenib-induced survival rates reduction (Supplementary Fig. 2c) and apoptosis (Supplementary Fig. 2d and 2e).

4. The authors found a dissipation of MMP. How can they explain this result? Is it due to decreased mitochondrial respiration, decreased mitochondrial mass? The authors should measure one of these parameter.

Response: Thanks for your advice. The loss of MMP is mainly dependent on the disruption of mitochondrial membrane and the formation of mitochondrial permeability transition pore (MPTP). Here we found that in regorafenib-induced hepatotoxicity, p53 was involved in regulating mitochondrial dysfunction. When stress stimuli present, p53 will translocate into mitochondrial outer membrane and binds to anti-apoptotic proteins, such as Bcl-2 or Bcl-xL, leading into the release of Bax and Bak. Bax and Bak can form the permeability pore respectively, which mediate the mitochondrial dysfunction

through the formation of MPTP. p53 also can transactivate the target gene PUMA which would fascinate the activation of apoptotic proteins and induce the formation of MPTP (*Tumour Biol.* 2016 Jul;37(7):8471-86.). When we silencing p53 in hepatocytes, regorafenib-induced MMP loss could be greatly reversed, suggesting the dominant role of p53 in regulating regorafenib-induced mitochondrial dysfunction. As previously reported, regorafenib could inhibit the respiratory chain and increase the mitochondrial production of superoxide which would further influent the MMP (*J Appl Toxicol.* 2018 Mar;38(3):418-431. and *Toxicology.* 2018 Feb 15;395:34-44.). We used mitoTracker to detect the mitochondrial mass and detected the ATP production after regorafenib treatment. Regorafenib could decrease the mitochondrial mass and ATP production in a dose-dependent manner (Fig.1f and 1g). When applied with siRNA targeting p53, EphA2 and Schisandrin C, we found that these interventions could greatly alleviate the loss of mitochondrial mass and ATP production (Fig.2g, 2h, 3f, 3g, S18c and S18d), suggesting EphA2 and p53 were acted as an accelerator of regorafenib-induced mitochondrial dysfunction.

Fig.1f**Fig.1g****Fig.3f****Fig.3g****Fig.3f****Fig.3g****Fig.S18c****Fig.S18d**
Newly released figures

5. It is not clear whether the effect on mitochondria is an on or off-target effect?

What is exactly the link to EphA2. Please discuss.

Response: Thank you for the comment. As we discussed in last response, the effect of regorafenib on mitochondria is partly dependent on the off-target effect. When treating isolated mitochondrial, regorafenib could slightly decrease MMP at onefold Cmax (~8.08 μM) (*Arch Toxicol. 2017 Aug;91(8):2921-2938.*). In general, EphA2 inhibition

may affect the signaling pathway of PI3K/AKT, AMPK and Ras/MAPK, and derive related phenotypes (*Biomed Res Int.* 2018 Feb 28;2018:7390104.). In our study, we found the link between EphA2 inhibition and MMP loss was mediated by stabilizing p53 which was not reported before.

6. The authors found that regorafenib induces apoptosis. This result is not so novel. Please provide the relevant references.

Response: Thanks for your suggestion. It is truly that regorafenib induces apoptosis by inhibiting related signaling pathway in several cancers, including bladder carcinoma, hepatocellular carcinoma, colorectal tumor, etc (*Anticancer Res.* 2017 Sep;37(9):4919-4926., *Oncol Rep.* 2017 Feb;37(2):1036-1044., *Clin Cancer Res.* 2014 Nov 15;20(22):5768-76. and *Clin Cancer Res.* 2014 Jul 1;20(13):3472-84.). For hepatotoxicity, regorafenib was reported to induce apoptosis in human hepatocyte cell lines (*J Appl Toxicol.* 2018 Mar;38(3):418-431. and *Toxicology.* 2018 Feb 15;395:34-44.). Relevant references are cited in the manuscript.

Reviewer #2 (Remarks to the Author):

This manuscript by Yan and colleagues assesses the mechanism of regorafenib-induced liver toxicity. Briefly, they provide evidence that regorafenib induces hepatocyte death and mitochondrial dysfunction, induce p53 protein stability, and suppress phosphorylation of EphA2, ERK1/2, and MDM2. The authors utilized hepatocyte cell lines, human primary hepatocytes, and immunocompetent mice to assess the toxicity of regorafenib. This is an interesting manuscript that could have significant implications for the field. The work presented is sound and rigorous, but most evidence is correlative and key studies are lacking to fully elucidate the mechanism. Additionally, the conclusions drawn from some key experiments exhibit some flaws. These significant concerns and others are outlined below and should be addressed.

1. In Figure 2e-f, it is suggested that p53 is driving cell death in regorafenib-treated hepatocytes. Knockdown of p53 did not rescue the apoptosis induced by regorafenib, only modestly decreasing cell death detected by PI/Annexin-V staining. The same is true for survival (Figure 2c). However, a p53 knockdown resulted in a more substantial recovery of mitochondrial membrane potential (JC-1 staining) and PARP cleavage. Likewise, EphA2 knockdown decreases PARP-cleavage and p53 levels (Figure 3c) but had only a modest impact on PI/Annexin-V cell death (Supplementary Figure 5d) or even cell survival (Figure 3b). Regorafenib was found to induce cell death in colorectal cancer cells via PUMA that was not dependent on p53 mutational status (Chen D, et al. *Clin Cancer Res.* 20(13):3472-3484. 2015.) Do EphA2 and p53 rescue mitochondrial function and not cell death?

Response: Thanks for your comment. PUMA is one of the Bcl-2 family members and can be transcriptionally activated in a p53 dependent or independent manner. As previously reported, regorafenib could increase the level of PUMA in a dose and time dependent manner in colorectal cancer cells (*Clin Cancer Res.* 2014 Jul 1;20(13):3472-84.). We treated hepatocytes with regorafenib and detected the expression of PUMA. The data was shown in Fig.R2. PUMA was not increased after regorafenib treatment which suggested that PUMA may not be involve in regorafenib-induced hepatocytes apoptosis. Regarding the obvious reversal of mitochondrial membrane potential and the insignificant reversal of survival and the level of c-PARP, it may be due to the different treatment period. The period of regorafenib treatment in JC-1 staining was 24 h and for PI/Annexin V staining and cell survival was 48 h. Short period of regorafenib-treatment was applied to detect the survival and apoptotic rates. We applied p53 knockdown or EphA2 knockdown hepatocytes treated with regorafenib for 36 h, the survival rates were greatly improved (Fig.2c and 3b) and the apoptotic rates were reduced from $35.8 \pm 3.0\%$ to $9.5 \pm 3.2\%$ and $7.6 \pm 1.4\%$ (Fig.2e), $30.5 \pm 4.5\%$ to $12.0 \pm 5\%$ and $9.9 \pm 3.4\%$ (Fig.3d), respectively. The results of JC-1staining for 24 h were also shown in Fig.2f and Fig.3e.

Editorial Note: The figure below has been redacted to remove third-party material where no permission to publish could be obtained.

[redacted]

Fig.R2

Fig.2c

Fig.2e

Fig.2f

Fig.3b

Fig.3d

Fig.3e

Newly released figures

2. If EphA2 S897 phosphorylation was the key driver in p53 upregulation and cell death, then expression of EphA2-S897A should lead to increased p53 protein even in the absence of regorafenib (Supplementary Figure 9). However, regorafenib was still required to increase p53 protein. Is EphA2 kinase activity required to stabilize p53?

Response: Thanks for your question. In old Fig.S9, we put on a blot of p53 at a short exposure which was not quite noticeable changed. In this version, we added a long exposure band to reflect EphA2-S897A could slightly increase p53 in the absence of regorafenib (Fig.4g). We also applied EphA2 loss of function mutation K646M to test whether EphA2 kinase activity was required for stabilizing p53. EphA2-K646M would block the kinase activity when phosphorylated EphA2 at Tyr588 (*J Biol Chem. 2005*

Dev Cell. 23;280(51):42375-82.). EphA2-K646M seems not affect the downstream regulation activity of p-EphA2 (Ser897) (*Dev Cell.* 2020 Aug 10;54(3):302-316.e7.). When we overexpressed EphA2-K646M, regorafenib could still increase the level of p53 (Fig.S11). Otherwise, we found that EphA2 (Ser897) inhibition was related to the stability of p53 both in ALW-II-41-27 and bosutinib treatment comparing with other EphA2 inhibitors (Fig.5f and Fig.S13).

Fig.S11

Fig.5f

Fig.S13a

Fig.S13b

Fig.S13c

Fig.S13d

Newly released figures

3. Ser897 phosphorylation of EphA2 is indicative of ligand-independent activation, while Tyr588 phosphorylation is indicative of ligand-dependent activation. How does ephrin-A1 impact the proposed signaling pathway? Does regorafenib-induced toxicity change ephrin-A1 expression in the liver?

Response: Thanks for your question. We treated hepatocytes with regorafenib and EFNA1 (also known as ephrin A1) and detected the impact of EFNA1 on regorafenib-induced hepatotoxicity. As shown in Fig.S10, we noticed that with regorafenib treatment solely, the expression of EFNA1 was greatly reduced which may explain the effect of regorafenib on the inhibition of Tyr588 phosphorylation (Fig.S10a). Silencing or overexpression of EFNA1 couldn't reverse regorafenib's effect on the level of p53

(Fig.S10b and S10c). After EFNA1 treatment, Tyr 588 was phosphorylated which was consisted with the reference reported. However, there is little influence on the survival rate (Fig.S10d), the expression of p53 and the proposed downstream p-ERK (Fig.S10e). These results may reflect that EFNA1-mediated EphA2 activation was not participate in the regulation of p53 stability.

Newly released figures

4. The data are not sufficient to support the mechanism of ERK1/2 on MDM2 phosphorylation. Regorafenib decreases ERK1/2 phosphorylation, but this may be due to inhibition EphA2 or other receptor tyrosine kinases. Data shown in Supplementary Figure 15 is proposed to show that ERK1/2 phosphorylation is associated with EphA2 S897 phosphorylation status. However, the differences are quite weak and based only on a single cell with no quantitation shown. Additionally, ERK has also been shown to regulate EphA2 S897 phosphorylation (Hamaoka Y, et al. Cell Signal. 28(8):937-945. 2016.), indicating that ERK may be upstream of EphA2. Please address.

Response: Thanks for pointing this out. As previously reported, AKT-mTOR, ERK pathways could be activated through ligand-independent activation of EphA2 (phosphorylation of Ser897) (*Hepatology. 2013 Jun;57(6):2248-60.*), and RSK, AKT, PKA, ERK and et la could also regulate the phosphorylation of EphA2 S897 depending

on different stimuli (*Biol Pharm Bull.* 2017;40(10):1616-1624.). The relation between the up/down-stream of EphA2 is still complex. In this study, we wanted to testify under regorafenib treatment, ERK or partial ERK could be regulated by EphA2. When knocking down EphA2, the effect of p-ERK inhibition could be greatly alleviated, which suggested EphA2 was upstream of ERK under regorafenib treatment (Fig.S16a). Due to obvious inhibition of phosphorylation of EphA2 at S897 in a short period, we shortened the actuation duration of regorafenib and tried to distinguish the changes in order. We added densitometric analysis to digitalize the difference. From the data, we could find out that p-EphA2 inhibition was much sensitive to regorafenib stimuli than that of p-ERK, both in response time and degree of inhibition (Fig.5h). Otherwise, we applied ERK activator TBHQ to treat with regorafenib (*J Biol Chem.* 1997 Nov 14;272(46):28962-70.). As shown in Fig.S16b, TBHQ could activate ERK and slightly increase the level of p-EphA2 (Ser 897) while TBHQ couldn't recover the inhibition of p-EphA2 when combined with regorafenib. Regorafenib could still inhibit the phosphorylation of EphA2 and reduced the level of p-ERK with TBHQ treatment to some extent. In this way, we confirmed EphA2 is the upstream of ERK at least in regorafenib-treated hepatocytes.

Fig.5h

Fig.S16a

Fig.S16b

5. In several blots, regorafenib appears to modestly reduce the protein levels of EphA2 (for example, Figure 3c, Figure 4b). Are expression levels of EphA2 reduced in regorafenib-treated cells? EphA2 phosphorylation should be normalized to total EphA2 and not GAPDH.

Response: Thanks for your suggestion. We replaced the densitometric analysis with p-EphA2/EphA2. From the data, we could still consider that p-EphA2 was inhibited even though the level of EphA2 was slightly reduced under regorafenib treatment. The data were shown in newly released Fig.4c. Because the activation of ERK has been reported to promote EphA2 expression (*Cancer Cell. 2005 Aug;8(2):111-8.*), it is possible that the slight downregulation of EphA2 is the negative feedback of p-ERK inhibition.

Newly released figures

6. On lines 320-321, the authors indicated that regorafenib did not impact the distribution of EphA2, but this claim is not substantiated by any evidence. In fact, in Supplementary Figure 7b it appears that EphA2 may be more localized to the

plasma membrane or at cell-to-cell interactions in regorafenib-treated hepatocytes.

Please address.

Response: Thanks for this comment. In the old version, we wanted to express that EphA2 didn't translocate to the cytoplasm or even the nuclei so as not to directly affect MDM2 or p53. We deleted related description in the new version and added "As previously reported, AKT-mTOR, ERK pathways could be activated through ligand-independent activation of EphA2 (phosphorylation of Ser897)" to connect the context. For EphA2 may localize to plasma membrane under regorafenib treatment, we considered that EphA2 may form dimer instead of the increment of EphA2 anchoring due to the slight reduction of EphA2. It is reported that EphA2 prefers the dimer formation in the absence of the ligand ephrin A1. This dimerization suppresses the phosphorylation of EphA2 at Ser897 and attenuates the function of Ser897 in comparison with the EphA2 monomer (*J Biol Chem.* 2015 Nov 6;290(45):27271-27279.). We could also observe that regorafenib greatly reduces the level of ephrin A1 (Fig.S10a), which may induce the preference of EphA2 to form dimer under regorafenib and make it more observable on plasma membrane by immunofluorescence.

Fig.S10a

Newly released figures

7. In Figure 6, MDM2 is significantly lower in the nuclear fraction of AAV-EphA2-898A livers but is unchanged in the WCL. This discrepancy does not appear to be associated with an increase in the cytoplasmic fraction, which is suggested in Supplementary Figure 15. This is also true in regorafenib-treated cells and livers shown in Figure 5b. If not the cytoplasm or degradation, where is MDM2 going?

Response: Thanks for pointing out this problem. The lower fraction of cytoplasmic may originate to the difference of sample loading. In the last version, the protein loading amount of nuclear fraction is 10 times larger than that of cytoplasmic fraction. We

readjust the proportion and separate each fraction for individual detection and quantification (Fig.5c and 6f).

Fig.5c

Fig.6f

Newly released figures

8. The knockdown of EphA2 in Figure 3 is quite poor, making it surprising to see complete recovery of p53 and c-PARP. In addition, additional EphA2 siRNAs should be used to increase confidence that these impacts on c-PARP and p53 are not due to off-target effects.

Response: Thanks for your advice. We applied new siRNA targeting EphA2, and used two different sequence to detect the effect of silencing EphA2 on the survival rate of regorafenib-treated hepatocytes and the level of c-PARP and p53. Both EphA2 siRNAs-treated hepatocytes showed similar response to regorafenib. Related results were shown in newly released Fig.3b and 3c.

Fig.3b

Fig.3c

Newly released figures

9. The authors suggest liver morphology changes, including nuclear size and immune infiltration, occur in mice treated with regorafenib (Figure 1a), infected with AAV (Figures 6b, 7c), and treated with regorafenib +/- Schisandrin C (Figure 8c). These observations, particularly those made on AAV models, are not obvious differences compared to control livers. Were these observations made in collaboration with a pathologist?

Response: Thank you for pointing this out. First, we noticed that after several condensation by PDF, the resolution of the figure is quite low and is difficult to distinguish the differences. We tried to improve the output of the manuscript. For HE staining, we put a new picture of a larger area and these changes can be observed in the global observation and replace some of the representative images to make the cohesive among different animal experiments. Liver morphology changes were marked in black arrowhead and liver injury mainly characterized as nuclear shrinkage and structure disorder with vacuolization.

Newly released figures

10. On lines 385-386, it was indicated that a drug screen was used to identify Schisandrin C as a compound that increases EphA2 phosphorylation. Please show the data from the screen.

Response: We used 45 natural products (usually used and other projects related) and 41 approved drugs (without obvious hepatotoxicity). For initial screening, the concentration used for natural products was 40 μ M and 10 μ M for approved drugs. Western blot was used to detect the ratio of p-EphA2 (Ser897)/EphA2 (Fig.S17a and S17b). The ratio larger than 3.0 was selected, and these compounds were further combined with regorafenib (Fig.S17c). These compounds contain Aripiprazole, Schisandrin C and Dioscin. Further, the survival rate was detected when combined with regorafenib (Fig.S17d). The most effective compound Schisandrin C was screened out.

Fig.S17a**Fig.S17b****Fig.S17c****Fig.S17d**
Newly released figures

11. Does Schisandrin C change phosphorylation of other EphA2 sites? If the EphA2 kinase domain is mutated, does this abrogate the effects of Schisandrin C?

Response: In newly released Fig.R3, after Schisandrin C treatment, it has little effect on the phosphorylation of EphA2 Tyr588 while increase the level of p-EphA2 (Ser897) (Fig.R3). When the kinase domain is mutated (for the downstream regulation of p-EphA2 (Tyr588) (*J Biol Chem.* 2005 Dec 23;280(51):42375-82. and *Dev Cell.* 2020 Aug 10;54(3):302-316.e7.), Schisandrin C could still activate p-EphA2 (Ser897) and reduce the level of p53 (Fig.S19c), which further suggested that Schisandrin C regulated p53 by activating p-EphA2 (Ser897) instead of p-EphA2 (Tyr588).

Fig. R3**Fig.S19c**
Newly released figures

12. Can the authors address why Schisandrin C rescues regorafenib-induced liver toxicity (Supplementary Figure 16) but has no impact on regorafenib efficacy on hepatocellular carcinoma (Supplementary Figure 19)?

Response: As previously reported, regorafenib, as a multikinase inhibitor, inhibits several oncogenic receptor tyrosine kinases, such as KIT, RET, RAF to have potent preclinical antitumor activity and EphA2 may not be a dominant target for regorafenib efficacy (*Int J Cancer. 2011 Jul 1;129(1):245-55.*). To date, the role of EphA2 in HCC has not been well explored. Several studies suggested that expression of EphA2 and its ligand correlated to the tumor progression, invasion, metastasis, and poor prognosis in patients with HCC (*Int J Cancer. 2010 Feb 15;126(4):940-9. And Hepatol Res. 2009 Dec;39(12):1169-77.*). On binding the ligand, Try588 of EphA2 was auto-phosphorylated. The EphA2 inhibitor ALW-II-41-27 could effectively decrease the phosphorylation of EphA2 at Try588 and its downstream effectors in HCC cells, and thus has an impact on HCC tumor growth (*Cell Rep. 2021 Feb 23;34(8):108765.*). In another study, the authors reported that Ephrin A3/EphA2 axis is a hypoxia-sensitive modulator of cancer stemness in HCC cells. Ephrin A3 regulates the phosphorylation of EphA2 at Try588 to trigger the metabolism reprogramming in HCC cells (*J Hepatol. 2022 Aug;77(2):383-396.*). Thus, it seems that the phosphorylation of EphA2 at Try588 is crucial for HCC growth and progression. In this study, we found that the phosphorylation inhibition of EphA2 at Ser897 was the key event of regorafenib-induced hepatotoxicity and we also proved that Schisandrin C didn't affect the level of phosphorylation of EphA2 at Tyr588, so this may be the reason that Schisandrin C

rescues regorafenib-induced hepatotoxicity but has no impact on regorafenib efficacy on hepatocellular carcinoma.

13. It is unclear why no statistical comparisons are not shown between samples treated with vehicle or regorafenib in siRNA/EphA2 mutant experiments. For example, in Figure 2e apoptosis is still quite elevated in p53-knockdown cells treated with regorafenib compared to those treated with vehicle. This occurs throughout the manuscript and can have a significant impact on data interpretation. Please include these statistical comparisons.

Response: Thanks for pointing out this issue. We checked the manuscript and included missed statistical comparisons. All the histograms were shown with scatters.

Fig.2c**Fig.2d****Fig.2e****Fig.2h****Fig.3b****Fig.3c****Fig.3g****Fig.3h****Fig.3d****Fig.3g****Fig.3h****Fig.3i****Fig.3j****Fig.4g****Fig.4h****Fig.7d****Fig.7e**
Newly released figures

14. Several of the blots and images shown do not appear to be representative of the corresponding quantitated data. In Supplementary Figure 2f, the bar graph shows the c-PARP in Regorafenib + Z-VAD-FMK is still considerably higher than

the Z-VAD-FMK alone lane, but there is no difference in the blot. In the EphA2 knockdown lysates shown in Figure 3c, cleaved PARP and p53 appears to be higher in the regorafenib lane than the untreated lane, but the combined data shows no difference. Showing values below the lane of each representative blots and individual data points on bar graphs for all plots may help.

Response: Thanks for pointing out this issue. We checked the raw data and added scatter to the plot.

Minor concerns:

15. Some of the fonts and scales, particularly on the flow cytometry plots, are impossible to read. Percentages for define populations should be indicated for representative data.

Response: Thanks for your advice. Because of the several condensations of the PDF, the resolution is quite low to distinguish the fonts and scales. We enlarged some of the fonts and tried to promote the resolution of each picture in the new version. We hoped that some of the key points could be readable now.

Fig.S2e

Fig.S8b

Fig.S8c

Newly released figures

16. In some JC-1 staining experiments, it is unclear why the gates are drawn as they are as there is not clear indications of separate populations existing. This is particularly true in Figure 1d, Figure 2f, Supplementary Figure 5e. Do the authors have supporting controls for these experiments?

Response: Thanks for your comment. In Fig. 1d, 2f and S5e, the gates were drawn due to the separation populations of each control. Sometimes, it is clear enough to roughly distinguish two population. Regorafenib has a strong impact on mitochondria which is confirmed by MitoTracker staining and ATP detection. With regorafenib treatment, it would be difficult to separate the two population with main group moved. In the new version, we added CCCP treatment as the positive control, and the gates were drawn dependent on the separation population between control and positive control. The representative image of Fig.2f and Fig.3e were shown below.

Fig.4Ra

Fig.4Rb

Newly released figures

17. No quantified data is shown for the TUNEL images shown in Figure 3e.

Response: Thanks for pointing out this issue. We have added quantification for TUNEL images throughout the manuscript, including newly released Fig.1h, 3i, 6c, 7d and 8d.

Newly released figures

Reviewer #3 (Remarks to the Author):

Comments:

Hepatotoxicity of regorafenib is one of the most noteworthy concerns for patients, however the mechanism is still unclear up to now. In this study, authors identified the inhibition of EphA2 Ser897 phosphorylation (human) as a key cause of regorafenib-induced hepatotoxicity, and chemical activation on EphA2 Ser897 represents a potential therapeutic strategy to prevent regorafenib-induced hepatotoxicity. This is a very interested story and very significant for clinical HCC treatment with regorafenib.

1. The opinion of manuscript is very interesting, however, the data is too rough and no enough critical result is for supporting. The antibodies were not explained using for mice or human, or both in the method. For example, the authors said the anti-c-PARP antibody information ((ET1608-10, Huabio, Hangzhou, Zhejiang,

China). However, the results showed that 25Kd cleaved band was detected in the mouse tissue (Fig 1f, no molecular weight information, we can find it in the primary data) and the band between 90kd to 100kd was detected in human cells and tissue. I think these results from different antibodies but the authors failed to show the antibodies information.

Response: Thanks for your kind advice. We have fulfilled the information in the antibody table including the reaction species and observed molecular weight (Supplementary Table.4).

Antibody	Company	Catalogue No.	Reactivity	Application used in this study	Observed Molecular weight (kDa)
GAPDH	Diagbio	db106	H, M	Western blot	37
P53	Santa Cruz Biotechnology	sc-126	H, M	Western blot; IHC staining	53
EphA2	Cell Signaling Technology	#6997	H, M	Western blot; IF	130
phospho-EphA2 (Ser897)	Cell Signaling Technology	#6347	H, M	Western blot; IF	130
MDM2	Huabio	RT1382	H, M	Western blot	55/100
MDM2	ABclonal	A0345	H, M	Western blot	55
phospho-MDM2 (Ser166)	Cell Signaling Technology	#3521	H, M	Western blot; IHC staining	90
Cleaved PARP	Huabio	ET1608-10	H	Western blot	89
Cleaved PARP	Abcam	ab32064	M	Western blot	25
HA tag	Diagbio	db2603	H, M	Western blot	N/A
FLAG tag	Diagbio	db7002	H, M	Western blot	N/A
Lamin B1	Huabio	R1508-1	H, M	Western blot	66
ERK	Santa Cruz Biotechnology	sc-81457	H, M	Western blot	42
LC3A/B	Cell Signaling Technology	#4108	H, M	Western blot	14, 16
phospho-Akt (Ser473)	Cell Signaling Technology	#4060	H, M	Western blot	65
P53	Origene	TA502870	H, M	IHC staining	N/A
MDM2	Affinity Biosciences LTD	AF0208	H, M	IF	N/A
phospho-EphA2 (Tyr588)	Cell Signaling Technology	#12677	H, M	Western blot	130
phospho-ERK (Tyr204)	Santa Cruz Biotechnology	sc-7383	H, M	Western blot; IHC staining; IF	42
EFNA1	ABclonal	A9132	H, M	Western blot;	25

2. All WB results are no molecular weight information.

Response: Thanks for pointing out this issue. We have added the molecular weight of protein marker around the observed band which is consistent with the raw data. You can find the alteration in the new version.

3. In addition, the EphA2 Ser898 or Ser897 phosphorylation in the abstract is also very confusing to reader, because the authors didn't say the EphA2 Ser898 from mouse EphA2 gene and Ser897 from human EphA2 gene.

Response: Thanks for pointing our mistakes. We have added additional description in the abstract part. "Ser898 for mouse EphA2" and "Ser897 for human EphA2" were added to distinguish the difference.

Abstract

Hepatotoxicity of regorafenib is one of the most noteworthy concerns for patients, however the mechanism has poorly understood. Hence, lack of effective intervention strategy. In this study, we found that regorafenib-induced liver injury mainly derived from its non-therapeutic target EPH receptor A2 (EphA2) by comparing the target with sorafenib. EphA2 deficiency attenuated liver damage and cell apoptosis under regorafenib treatment. Overexpression of EphA2-S898A (Ser898 for mouse EphA2) could directly cause hepatocyte apoptosis, while recovering EphA2 Ser898 phosphorylation greatly alleviated regorafenib-induced liver injury in male mice. Mechanistically, regorafenib inhibits EphA2 Ser897 (Ser897 for human EphA2) phosphorylation and reduced ubiquitination of p53 by altering the intracellular localization of mouse double minute 2 (MDM2) through affecting the extracellular signal-regulated kinase (ERK)/MDM2 axis. Meanwhile, we found Schisandrin C which could upregulate the phosphorylation of EphA2 at Ser897 also had protective effect against the toxicity *in vivo*. Collectively, our findings identify the inhibition of EphA2 Ser897 phosphorylation as a key cause of regorafenib-induced hepatotoxicity, and chemical activation on EphA2 Ser897 represents a potential therapeutic strategy to prevent regorafenib-induced hepatotoxicity.

REVIEWERS' COMMENTS

Reviewer #1 (Remarks to the Author):

The authors performed new experiments according my comments. The new results look solid and support the conclusion. I have no further comments to the authors.
I would recommend the publication of this study in "Nature Communications".

Reviewer #2 (Remarks to the Author):

In this revised manuscript, the authors have thoroughly addressed the reviewers comments. A few minor issues remain.

1. The authors have provided new convincing evidence on the EFNA1 ligand, but these new findings are not clearly addressed in the text (page 14). It would greatly improve the reader's understanding of EFNA1-EphA2 signaling if this related text was clarified.
2. A citation on ligand-dependent EphA2-Y588 phosphorylation (page 14) is missing.
3. SE and LE are not defined in the figure legend of Supplementary Figure 16.

Reviewers' Comments:

Point-by-point response to the reviewers' comments:

Reviewer #1 (Remarks to the Author):

The authors performed new experiments according my comments. The new results look solid and support the conclusion. I have no further comments to the authors.

I would recommend the publication of this study in "Nature Communications".

Response: We really appreciate your careful review and constructive suggestions with regard to our manuscript.

Reviewer #2 (Remarks to the Author):

In this revised manuscript, the authors have thoroughly addressed the reviewers comments. A few minor issues remain.

1. The authors have provided new convincing evidence on the EFNA1 ligand, but these new findings are not clearly addressed in the text (page 14). It would greatly improve the reader's understanding of EFNA1-EphA2 signaling if this related text was clarified.

Response: Thanks for your advice. We added the background of EFNA1-EphA2 signalling and rearranged the paragraph to make it easier to understand the purpose of the experimental design.

2. A citation on ligand-dependent EphA2-Y588 phosphorylation (page 14) is missing.

Response: Thanks for your comment. Related references were cited at the appropriate place on page 14 (*Biol Pharm Bull.* 2017;40(10):1616-1624. and *J Biol Chem.* 2008

Jun 6;283(23):16017-26.

3. SE and LE are not defined in the figure legend of Supplementary Figure 16.

Response: Thanks for pointing this out. We have added related missing abbreviations.

Author's note: In the last version of the point-to-point response, we find a small mistake in the figure for the Reviewer 1's fourth question. "Fig 3f" and "Fig 3g" were mistakenly copied.

For the MitoTracker staining and ATP content assay in p53-knockdown cells, the correct labels should be "Fig 2g" and "Fig 2h" respectively.